# Breast cancer metastasis suppressor OTUD1 deubiquitinates SMAD7

Zhengkui Zhang[1], Yao Fan[1], Feng Xie[2], Hang Zhou[1], Ke Jin[1], Li Shao[3], Wenhao Shi[4,5,6], Pengfei Fang[1], Bing Yang[7], Hans van Dam[8], Peter ten Dijke[8], Xiaofeng Zheng[9], Xiaohua Yan[10], Junling Jia[1], Min Zheng[3], Jin Jin[1], Chen Ding[4,5,6], Sheng Ye[1], Fangfang Zhou [2] & Long Zhang [1]

Metastasis is the main cause of death in cancer patients. TGF-β is pro-metastatic for malignant cancer cells. Here we report a loss-of-function screen in mice with metastasis as readout and identify OTUD1 as a metastasis-repressing factor. OTUD1-silenced cancer cells show mesenchymal and stem-cell-like characteristics. Further investigation reveals that OTUD1 directly deubiquitinates the TGF-β pathway inhibitor SMAD7 and prevents its degradation. Moreover, OTUD1 cleaves Lysine 33-linked poly-ubiquitin chains of SMAD7 Lysine 220, which exposes the SMAD7 PY motif, enabling SMURF2 binding and subsequent TβRI turnover at the cell surface. Importantly, *OTUD1* is lost in multiple types of human cancers and loss of OTUD1 increases metastasis in intracardial xenograft and orthotopic transplantation models, and correlates with poor prognosis among breast cancer patients. High levels of OTUD1 inhibit cancer stemness and shut off metastasis. Thus, OTUD1 represses breast cancer metastasis by mitigating TGF-β-induced pro-oncogenic responses via deubiquitination of SMAD7.

[1] Life Sciences Institute and Innovation Center for Cell Signalling Network, Zhejiang University, Hangzhou, 310058 Zhejiang, China. [2] Institutes of Biology and Medical Science, Soochow University, 215123 Suzhou, China. [3] State Key Laboratory for Diagnostic and Treatment of Infectious Diseases, The First Affiliated Hospital, School of Medicine, Zhejiang University, Collaborative Innovation Center for Diagnosis and Treatment of Infectious Disease, 310000 Hangzhou, China. [4] State Key Laboratory of Proteomics, Beijing Proteome Research Center, Beijing Institute of Radiation Medicine, 100039 Beijing, China. [5] National Center for Protein Sciences (The PHOENIX Center, Beijing), 102206 Beijing, China. [6] State Key Laboratory of Genetic Engineering and Collaborative Innovation Center for Genetics and Development, School of Life Sciences, Institute of Biomedical Sciences, Fudan University, 200433 Shanghai, China. [7] Department of Pharmaceutical Chemistry and the Cardiovascular Research Institute, University of California San Francisco, San Francisco, CA 94158, USA. [8] Department of Molecular Cell Biology and Cancer Genomics Centre Netherlands, Leiden University Medical Center, Postbus 9600, 2300 RC Leiden, The Netherlands. [9] Department of Cancer Biology, Metastasis Research Center, University of Texas MD Anderson Cancer Center, Houston, TX 77054, USA. [10] Tsinghua University, 100084 Beijing, China. Zhengkui Zhang, Yao Fan and Feng Xie contributed equally to this work. Correspondence and requests for materials should be addressed to L.Z. (email: L_Zhang@zju.edu.cn)

Metastatic disease is largely incurable because of its systemic nature and the resistance of disseminated tumor cells to existing therapeutic agents[1]. To colonize distant organs, circulating tumor cells must overcome many obstacles, including surviving in circulation, infiltrating distant tissues, evading immune defenses, adapting to supportive niches, surviving as latent tumor-initiating seeds, and eventually breaking out to replace the host tissue[2]. Metastasis is a highly inefficient process and the mechanisms are poorly understood. TGF-β signaling is one of the most important pathways involved in all these metastatic processes[3–5]. In many late-stage tumors, TGF-β signaling is redirected from suppressing cell proliferation and instead found to activate epithelial-to-mesenchymal transition (EMT), a cellular program that promotes cancer cell intravasation and confers cancer stem cells traits associated with high-grade malignancy[6–8].

TGF-β signals via specific complexes of type I (TβRI) and type II Ser/Thr kinase receptors. The activated TGF-β type I receptor induces SMAD2/3 phosphorylation; phosphorylated SMAD2/3 forms hetero-oligomers with SMAD4, which accumulate in the nucleus to regulate the expression of target genes[9]. SMAD7 functions as an inhibitory SMAD by recruiting the E3-ubiquitin ligase SMURF2 to TβRI and mitigating TGF-β signaling[10–12]. Various E3 ligases, including ARKADIA and RNF12 can potentiate TGF-β signaling by targeting SMAD7 for poly-ubiquitination and degradation[13–16].

Recently, we developed an in vivo screen in mice that enables the isolation of genetic entities involved in activation of breast cancer metastasis. Here, the results of one such screen using a DUB shRNA library is presented. The top hit, termed OTU domain-containing protein 1 (OTUD1), was found to inhibit breast cancer stem cell traits and metastasis. We also elucidate the underlying mechanism and show that OTUD1 empowers SMAD7 to inhibit TGF-β signaling in breast cancer metastasis.

## Results

### Genetic screen identified OTUD1 as a metastasis suppressor.
We designed a loss-of-function screen in mice to identify deubiquitinating enzymes (DUBs) that antagonize metastasis (Fig. 1a; Supplementary Fig. 1a) and applied it to early passage MDA-MB-231 cells, which still show epithelial-like morphology and exhibit relatively low metastatic ability. We used a shRNA library targeting 74 DUBs, in which each DUB is covered by 4–6 independent short hairpins with at least two of them validated (Supplementary Data 1). Instead of making a pool of shRNA virus, we produced up to 371 distinct shRNA lentiviruses in HEK293T cells and individually introduced them into early passage MDA-MB-231 cells. After puromycin selection for three days, this gave rise to 371 stable cell lines. We used an equal amount of cells from each cell line ($10 \times 10^3$ cells per shRNA stable cell line) and mixed them for nude mice intracardiac injection (Supplementary Fig. 1a). Within 4 weeks, the mixed shRNA stable cells produced a total of seven strong metastatic nodules in multiple mice (3 from 30 mice shown in Fig. 1b); some of the other mice developed weak micrometastasis (6 from 30 mice shown in Fig. 1b). In contrast, cells infected with empty vector did not produce macroscopic lesions upon injection in 30 mice after 4 weeks (Fig. 1b). This screening strategy can thus be used to identify essential DUBs that suppress metastasis.

After examination and reintroduction in the early passage MDA-MB-231 cells, two of the seven shRNAs isolated from individual lesions promoted lung metastasis without affecting primary tumor growth. We focused on these two because both shRNAs target OTUD1 (Fig. 1b), an OTU domain family member with a largely unexploited function in cancer. OTUD1

was reported to be differentially expressed in thyroid cancer and to deubiquitinate and stabilize p53[17,18]. To consolidate our observations, we generated OTUD1 knockout cell lines of the highly bone-metastatic MDA-MB-231 (BM) cells[19,20] by two independent guide RNAs of CRISPR/Cas9 technology (Fig. 1c). Mice injected with these two OTUD1 knockout clones developed larger bone metastases and had significantly shorter bone metastases-free survival periods (Fig. 1e), corroborating a strong anti-metastatic activity of endogenous OTUD1. To directly implicate OTUD1 in this process, we used a doxycycline-regulated promoter to initiate its expression in MDA-MB-231 (BM) cells 3 weeks after intracardiac injection (Fig. 1d, f). Although cell transplantation induced bone metastasis to similar levels around day 21 post-injection, expression of OTUD1 wild-type (wt) beginning at day 21 significantly suppressed metastasis around day 35, which also led to increased mice survival (Fig. 1f). Thus, induction of OTUD1 causes inhibition of metastatic outgrowth, which is not observed with its deubiquitinating enzyme inactive form CA (carrying a point mutation in one of the key cysteines of the catalytic domain), directly implicating OTUD1 as a critical DUB in metastasis suppression.

Given the role of OTUD1 in the regulation of metastasis, we investigated the possibility that OTUD1 might be a relevant factor in late-stage cancer. Oncomine expression analysis revealed that OTUD1 mRNA levels are frequently downregulated in human cancers[21–26] (Fig. 1g; Supplementary Fig. 1b). Using the NKI295 breast cancer database[27], we observed that low OTUD1 expression is associated with poor prognosis of distant metastasis-free survival in patients (Fig. 1h). This feature is apparently not limited to breast cancer; in The Cancer Genome Atlas (TCGA) database[28], bladder urothelial patients with lower OTUD1 expression had a shorter disease-free life expectancy than those with higher OTUD1 expression (Supplementary Fig. 2a), suggesting that OTUD1 inhibits metastatic relapse in patients. Finally, hypothesizing that OTUD1 has a signaling function, we compared NKI295 tumor microarray data of 54 OTUD1-high expressing patients and 64 OTUD1-low expressing patients and applied it to Gene Set Enrichment Analysis (GSEA)[29]. Intriguingly, we observed that the 70 core targets of tumor suppressor Breast Cancer 1 (BRCA1) are all significantly enriched in OTUD1-high patients (Fig. 1i; Supplementary Fig. 2b and Supplementary Data 2), strongly indicating that OTUD1 could support BRCA1 function. These results are consistent with a role for OTUD1 in restricting metastasis of human breast cancer.

### OTUD1 inhibits cancer stem cell traits.
We next examined the role of OTUD1 in breast cancer cell phenotypic behavior. Depletion of endogenous OTUD1 potentiated the capacity of RAS transformed MCF10A cells to form tumor organoids in 3D Matrigel but did not inhibit tumor cell survival and proliferation under standard 2D culture conditions (Fig. 2a, b). Conversely, lentiviral-mediated expression of wt OTUD1, but not the CA mutant, significantly reduced the number of tumor-like colonies (Fig. 2c). To investigate the ability of OTUD1 to regulate cancer stem cell activity, we performed mammosphere assays[30]. Although silencing of endogenous OTUD1 strongly increased the capacity of MCF10A (RAS) cells to form tumor spheres, ectopic expression of OTUD1-wt, but not the CA mutant form, greatly reduced this (Fig. 2d–f). Consistent results in both tumor organoids and mammosphere assays were obtained when using mouse 4T1 mammary carcinoma cells (Supplementary Fig. 2c, d). These results indicate that OTUD1 inhibits stemness of breast cancer cells.

We subsequently examined the mitigating effect of OTUD1 on cancer stemness in vivo employing subcutaneous injections. Inactivation of OTUD1 promoted tumor incidence and tumor

growth and shortened latency after injection of limiting numbers of MCF10A-RAS cells in mice (Fig. 2g, h). Statistical analysis confirmed that silencing of OTUD1 increases the frequency of tumor-initiating cells (Supplementary Table 1). Conversely, in the same limited cell number injection assay we observed that ectopic expression of OTUD1-wt, but not the CA mutant, inhibited tumor incidence and growth and prolonged latency after injection of $10^2$, $10^3$, $10^4$ but not $10^5$ cells (Fig. 2j, k). In line with this, the transcription of the pluripotency factors *NANOG*, *SOX2*, *OCT4*, and *TAZ* were significantly elevated in OTUD1-depleted tumors

and severely inhibited in tumors expressing wild-type OTUD1 (Fig. 2i, l). These results suggest that OTUD1 reduces the manifestation of cancer stem cell traits.

**OTUD1 causes suppression of TGF-β downstream signals and EMT**. Stem cell traits of breast carcinoma are associated with EMT, which requires interplay of multiple tumor-promoting pathways including TGF-β[8]. In breast cancer cells, TGF-β treatment strongly promoted tumor organoids and tumor spheres

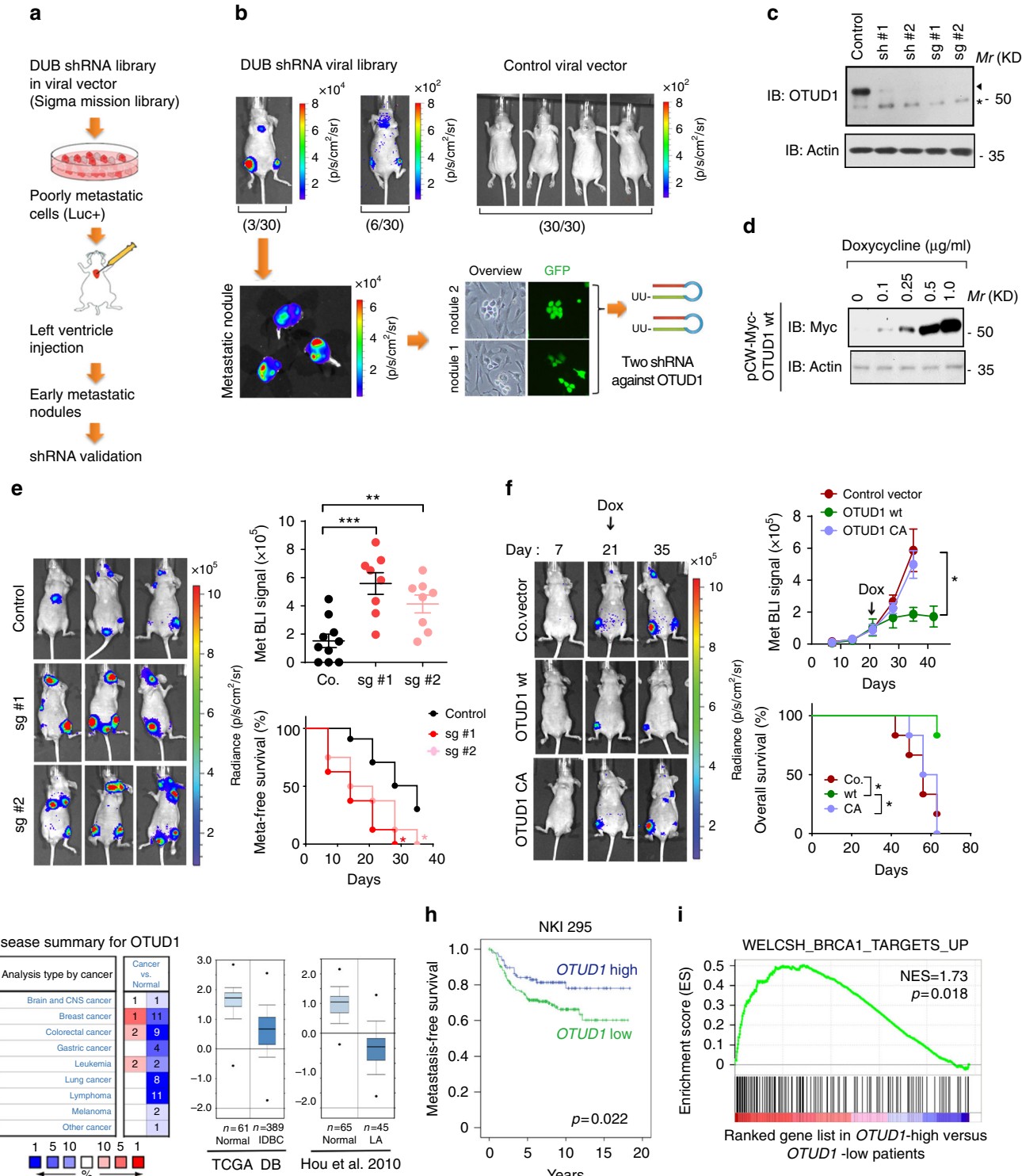

formation; and these effects were inhibited by TβRI kinase inhibitor SB431542 (Supplementary Fig. 3a, b), demonstrating that TGF-β signaling indeed play a positive role in supporting stem-cell-like properties. TGF-β signals via its downstream activation of SMAD2 and SMAD3[6], which can be measured with ARE-Luc and CAGA-Luc transcriptional reporters, respectively[31,32]. Among more than 70 tested DUB cDNAs, OTUD1 was identified as one of a few potent DUBs that antagonized both TGF-β induced ARE-Luc and CAGA-Luc reporters (Fig. 3a). OTUD1-CA did not inhibit TGF-β signaling (Fig. 3b), suggesting that DUB activity is essential. Knockdown assays demonstrated that loss of OTUD1 is required for a proper TGF-β-induced transcriptional response (Fig. 3c). Ectopic expression of OTUD1-wt, but not OTUD1-CA, mitigated the magnitude and duration of TGF-β-induced SMAD2 phosphorylation and SMAD2–SMAD4 complex formation in MCF10A-RAS cells (Fig. 3d). Depletion of OTUD1 showed the opposite effect (Fig. 3e). These results indicate that OTUD1 is a potent antagonist of TGF-β/SMAD signaling.

Typical features of stemness-associated EMT include upregulation of N-cadherin, fibronectin, smooth muscle actin and vimentin, and downregulation of E-cadherin. Upon depletion of OTUD1, epithelial HaCaT keratinocytes gained mesenchymal features as analyzed by confocal-microscopy (Fig. 3f). This latter effect is inhibited by the treatment of SB431542, a selective TβRI kinase inhibitor, suggesting that the autocrine TGF-β is sufficient to promote EMT in OTUD1-depleted cells. In breast cancer cells, silencing OTUD1 promoted TGF-β-induced changes in EMT-marker expression, whereas ectopic expression of OTUD1-wt, but not OTUD1-CA, had the reverse effect (Fig. 3g; Supplementary Fig. 3a). In line with this, PCR array analyses confirmed that loss of OTUD1 led to several molecular features of mesenchymal cells, including the upregulation of key transcriptional inducers such as TWISTs, SNAILs, ZEBs, and other EMT-related targets such as COL1A1, SERPINE1 (Fig. 3h, i; Supplementary Data 3); We next analyzed the influence of OTUD1 overexpression on well-established EMT-related makers genes at 3 days (immediate response) and 12 days (delayed response) post-transduction of OTUD1 expressing lentiviruses in breast cancer MDA-MB-231 (BM) cells (Fig. 3j). As expected, OTUD1 overexpression was associated with an increased expression level of the epithelial maker CDH1 and decreased expression levels of the mesenchymal markers CDH2, FN1, and VIM (Fig. 3j). Reduced expression of several EMT-related transcription factors, including SNAIL, SLUG, ZEB1, and ZEB2, was observed at 12 days post-OTUD1 overexpression (Fig. 3j), showing a reversal of the mesenchymal phenotype of MDA-MB 231 (BM) cells. Besides, all of these genes

were suppressed by an increase of OTUD1-wt, but not by OTUD1-CA, either at the basal or TGF-β-induced level (Supplementary Fig. 3b; Supplementary Data 3). Taken together, these results indicate that OTUD1 is a critical and selective inhibitor of TGF-β/SMAD signaling and EMT.

**OTUD1 associates with SMAD7 and deubiquitinates SMAD7 in vitro.** We next investigated the molecular mechanism by which OTUD1 inhibits TGF-β/SMAD signaling. The inhibitory effect of OTUD1 on TGF-β-induced SMAD2 phosphorylation (Fig. 3d) indicates that OTUD1 acts on the receptor/SMAD level. As OTUD1's DUB activity is required, targeting of an inhibitor of TGF-β signaling, such as inhibitory SMAD7, appeared most likely. Ectopic expression of OTUD1 could suppress TGF-β signaling in control cells but barely showed effect in SMAD7-deficient cells; also knockdown of OTUD1 could promote TGF-β signaling in control cells but not in SMAD7-deficient cells (Fig. 4a; Supplementary Fig. 4a), suggesting that inhibitory SMAD7 might be the major target of OTUD1 in the TGF-β pathway. To consolidate this observation in vivo, we generated a SMAD7 knockout cell line of early passage MDA-MB-231 cells using CRISPR/Cas9 technology (Supplementary Fig. 4a). Mice injected with this SMAD7 knockout clone developed more rapid and stronger metastasis and had significantly shorter metastasis-free survival periods (Fig. 4b). Moreover, ectopic expression of OTUD1 significantly inhibited metastasis in control cells but did not show significant inhibitory effect in the SMAD7 knockout cells. This demonstrates that the anti-metastatic activity of OTUD1 is largely mediated via regulation of SMAD7. Consistent with this notion, in vitro purified SMAD7 protein was found to associate directly with OTUD1 (Fig. 4c). In cells, Myc-tagged OTUD1 strongly and specifically co-precipitated with SMAD7 (Fig. 4d). Moreover, endogenous OTUD1 was found to interact with SMAD7 in breast cancer cells (Fig. 4e).

Although OTUD1-wt-associated SMAD7 was not ubiquitinated, OTUD1-CA mutant-associated SMAD7 was highly ubiquitinated, with the SMAD7 poly-ubiquitination band presenting most prominently (Supplementary Fig. 4b). We then examined whether OTUD1 serves as a DUB for SMAD7. Firstly, Flag-tagged SMAD7 proteins were affinity purified, and their ubiquitination pattern was visualized by immunoblotting for HA-ubiquitin: poly-ubiquitination appeared as a major modification of SMAD7. To demonstrate that OTUD1 can directly deubiquitinate SMAD7, we performed in vitro deubiquitination assays. Purified OTUD1-wt, but not OTUD1-CA, removed poly-ubiquitin chains from SMAD7; when an optimal concentration of OTUD1 was applied for this assay, more than 50% of the poly-

---

**Fig. 1** An in vivo genetic screen identifies OTUD1 as potent suppressor of breast cancer metastasis. **a**, **b** Flow chart and figures of the in vivo screen identifying DUBs that inhibit breast cancer metastasis. Low metastatic MDA-MB-231-Luciferase/GFP breast cancer cells were infected with lentiviruses expressing DUB shRNAs and intracardially injected into nude mice. The mice were monitored for 4 weeks by in vivo bioluminescent imaging (BLI) and the early metastatic nodules were isolated and the corresponding shRNAs were identified by sequencing. See Supplementary Fig. 1a for details. **c** Immunoblot (IB) analysis of OTUD1 shRNA and sgRNA-mediated knockdown and knockout in MDA-MB-231 (BM) cells. **d** IB analysis of MDA-MB-231 (BM) cells stably expressing doxycycline (dox)-inducible OTUD1 and treated with the indicated doses of Dox. **e** Left panel: BLI of three representative mice injected with MDA-MB-231 control cells or cells deficient in OTUD1, Images were taken 4 weeks after injection. Two independent sgRNAs were used to generate OTUD1 knockout cells. Right upper panel: BLI signals of all mice in each experimental group at week 5. Right lower panel: The percentage of metastasis-free mice in each experimental group followed in time. **f** MDA-MB-231 cells expressing DOX-inducible OTUD1-wt/CA was intracardially injected in mice. DOX was administrated 21 days after inoculation of the cells. Metastasis was analyzed by BLI. Left panel: BLI of representative mice from each group at indicated days. Right upper panel: BLI signal of each experimental group followed in time. Right lower panel: The percentage of metastasis-free mice in each experimental group followed in time. **g** Oncomine database summary for OTUD1 expression in multiple cancers (left). Box plots of OTUD1 expression levels in invasive ductal breast carcinoma (IDBC) and lung adenocarcinoma (LA) compared with normal tissue (right). **h** Kaplan–Meier analysis of relapse-free survival of patients in publicly available breast cancer datasets (NKI 295). **i** BRCA1 target genes are enriched in OTUD1-high versus OTUD1-low expressing patients shown by pre-ranked gene-set enrichment analysis (GSEA). $*p < 0.05$, $**p < 0.01$ and $***p < 0.001$ (two-tailed Student's $t$-test (**e**, **f**), Log-rank test (**h**) or two-way analysis of variance (ANOVA) (**e**, **f**))

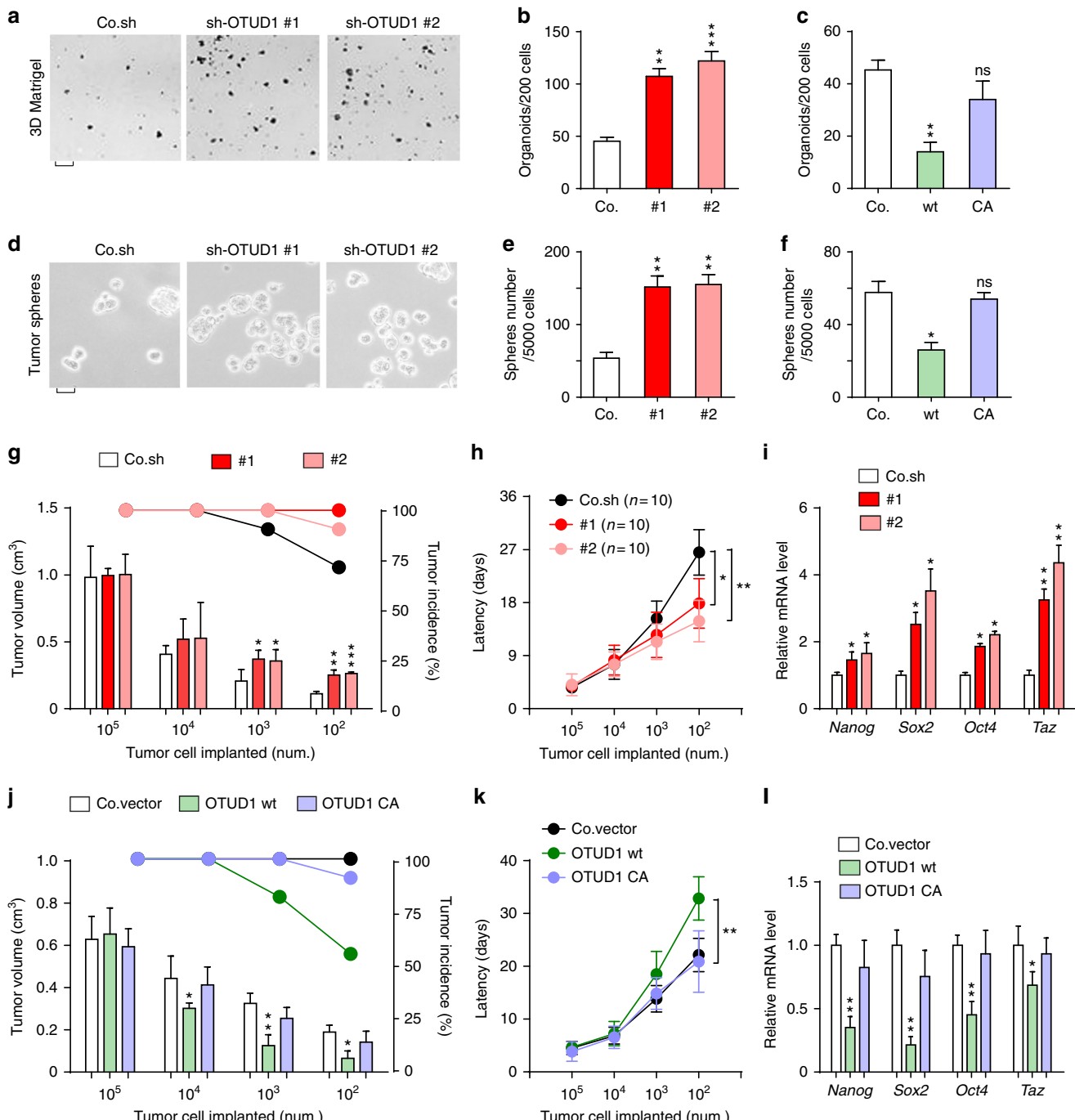

**Fig. 2** OTUD1 inhibits cancer stem cell traits. **a**, **b** Control and OTUD1-silenced MCF10A-RAS cells were cultured in 3D Matrigel. Representative wells **a** (scale bar, 2 mm) and mean number of organoids (±SE) per 200 cells from triplicate samples **b**. **c** Mean number of organoids (±SE) per 200 cells from triplicate samples of control and OTUD1-wt/CA overexpressed MCF10A-RAS cells cultured in 3D Matrigel. **d**, **e** Control and OTUD1-silenced MCF10A-RAS cells were analyzed in a tumor sphere assay. The pictures show representative images of tumor spheres (**d**, scale bar, 500 μm), and the graph shows the number of tumor spheres per $5 \times 10^3$ cells seeded **e**. **f** Mean number of tumor spheres per $5 \times 10^3$ cells from triplicate samples of control and OTUD1-wt/CA overexpressing MCF10A-RAS cells. **g** Control and OTUD1-shRNA silenced MCF10A-RAS cells were subcutaneous injected into nude mice at the indicated numbers. Mean of tumor volumes at week 5. **h** Tumor latency at indicated cell number of cells described in **g**. **i** Control and OTUD1-silenced MCF10A-RAS cells were subjected to qPCR analysis. **j** Control MCF10A-RAS cells or cells stably expressing OTUD1-wt or OTUD1-CA were subcutaneous injected into nude mice at the indicated numbers. Mean of tumor volumes at week 5. **k** Tumor latency at indicated cell number described in **j**. **l** Control MCF10A-RAS cells or cells stably expressing OTUD1-wt or OTUD1-CA were subjected to qPCR analysis. Error bars, mean ± SD. *$p < 0.05$, **$p < 0.01$, and *** $p < 0.001$ (two-tailed Student's $t$-test **b-l**)

ubiquitylated SMAD7 was cleaved within 20 min (Fig. 4f). When poly-ubiquitinated SMAD7 was purified with nickel beads from His-Ubiquitin expressing cells and applied to an in vitro deubiquitination assay, the poly-ubiquitin chains were deubiquitinated by OTUD1-wt. This results in a concomitant

accumulation of free SMAD7 (Fig. 4g), indicating that poly-ubiquitinated SMAD7 is a substrate of OTUD1. Previously, we have identified RNF12 as an E3 ligase for SMAD7[13]. Purified Gluthathion S-tranferase (GST)-RNF12 wt, but not GST-RNF12 CA mutant (the essential catalytic sites in C3HC4 motif of RNF12

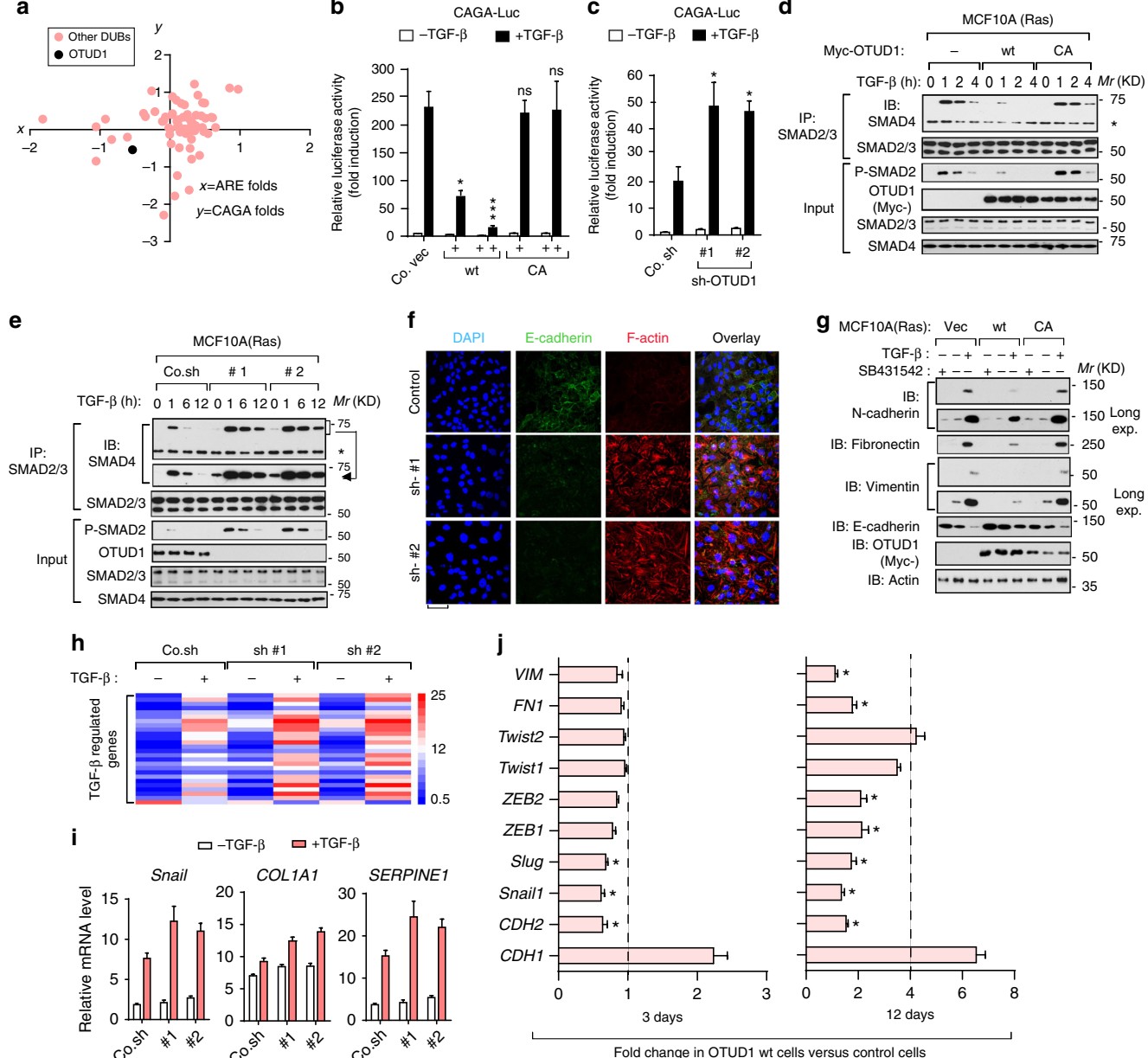

**Fig. 3** OTUD1 causes suppression of TGF-β downstream signals and EMT. **a** Diagram of DUB cDNA screen data in HEK293T cells in which DUBs that activate TGF-β-induced SMAD3/SMAD4-dependent CAGA$_{12}$-Luc transcriptional reporter and TGF-β-induced SMAD2/SMAD4-dependent ARE-Luc are indicated. The x-axe is the ARE-luciferase activity and y-axe is the CAGA-luciferase activity, shown as Log_2 value of ligand-induced fold changes. **b**, **c** Effect of ectopic OTUD1 wild-type (OTUD1-wt) and OTUD1-CA mutant (**b**) or OTUD1knockdown (sh-OTUD1 #1 and #2) (**c**) on CAGA$_{12}$-Luc transcriptional response induced by TGF-β in HEK293T cells. Co.vec, empty vector; Co.sh, non-targeting shRNA. **d**, **e** IB of total cell lysate and anti-SMAD2/3 immunoprecipitates derived from control and MCF10A-RAS cells stably depleted for OTUD1 (**d**) or stably expressing Myc-OTUD1-wt/C320A (**e**) and treated with TGF-β (5 ng/ml) as indicated. SMAD2/3-associated SMAD4 was analyzed by IB. 5% total cell lysate was loaded as input and were analyzed for P-SMAD2, OTUD1, SMAD2/3, and SMAD4. **f** Immunofluorescence and 4, 6-diamidino-2-phenylindole (DAPI) staining of control and HaCaT cells stably depleted of OTUD1 and treated with TGF-β (2.5 ng/ml) for 72 h. Scale bar, 50 μm. **g** IB of cell lysates derived from MCF10A(RAS) cells stably expressed with control empty vector (Vec), or OTUD1-wt/CA constructs and treated with TGF-β (2.5 ng/ml) and SB431542 (10 μM) for 72 h. **h** Heat map of TGF-β regulated genes in control cells (Co.sh) or MCF10A (RAS) cells stably depleted for OTUD1 with two independent shRNA (#1 and #2) and treated with or without TGF-β (2.5 ng/ml) for 8 h. **i** qRT-PCR analysis of TGF-β target genes Snail, COL1A1, and SERPINE1 in control and OTUD1 stably depleted MCF10A (RAS) cells treated with TGF-β (2.5 ng/ml) for 8 h. Values and error bars represent the means ± SD of triplicates and are representative of at least two independent experiments. **j** qRT-PCR analysis of MDA-MB-231 (BM) cells infected with control of OTUD1-wt-expressing virus for 3 days (left panel) or 12 days (right panel). *$p < 0.05$, **$p < 0.01$ and ***$p < 0.001$ (two-tailed Student's t-test **b–j**)

RING domain were mutated), greatly promoted SMAD7 poly-ubiquitination[13]. Upon incubation of His-SMAD7 with RNF12 in vitro, the poly-ubiquitin chains of SMAD7 were cleaved by OTUD1-wt (Fig. 4h). This confirms that OTUD1 deubiquitinates SMAD7 in vitro.

**OTUD1 deubiquitinates SMAD7 and sustains SMAD7 stability.** In cells we found overexpression of OTUD1-wt, but not OTUD1-CA, to deubiquitinate SMAD7 poly-ubiquitination including the Lysine 48 chain conjugation (as revealed by K48-linkage specific antibody) either in the absence

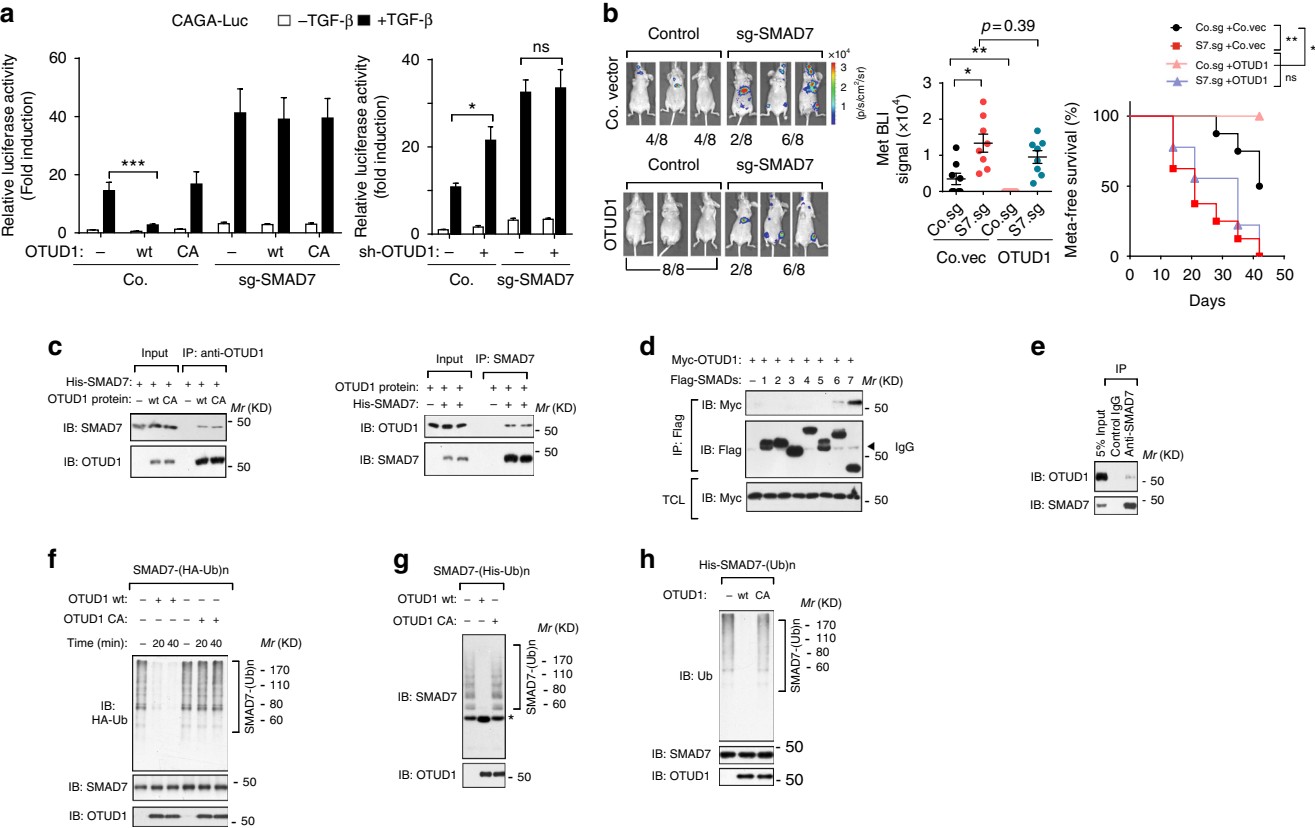

**Fig. 4** OTUD1 interacts with and deubiquitinates SMAD7 in vitro. **a** Effect of OTUD1-wt and CA mutant (left panel) or sh-OTUD1 #1 (right panel) on CAGA$_{12}$-Luc transcriptional response induced by TGF-β (2.5 ng/ml) for 16 h in HEK293T cells. Co., control sgRNA. **b** Left panel: BLI of three representative mice injected with MDA-MB-231 control cells or cells deficient in SMAD7 (sg-SMAD7) that ectopically expressed with control vector (Co.vec) or OTUD1, images were taken 6 weeks after injection. Middle panel: BLI signals of all mice in each experimental group at week 6. Right panel: the percentage of metastasis-free mice in each experimental group followed in time. **c** Purified SMAD7 and OTUD1-wt/CA interaction in vitro. Prokaryotic purified SMAD7 and eukaryotic purified OTUD1 proteins were incubated and immunoprecipitated with anti-OTUD1 (left panel) or with anti-SMAD7 (right panel) antibodies. Immunoprecipitates were then immunoblotted for OTUD1 and SMAD7. **d** IB analysis of total cell lysate (TCL) and immunoprecipitates derived from HEK293T cells transfected with Myc-OTUD1 and Flag-SMAD1-7 constructs as indicated. **e** IB of anti-SMAD7 immunoprecipitate derived from HEK293T cells. Association between OTUD1 and SMAD7 was examined. 5% total cell lysates was loaded as input. **f–h** OTUD1 deubiquitinates poly-ubiquitination of SMAD7 in vitro. Flag-SMAD7 and HA-Ub were transfected in HEK293T cells. Poly-ubiquitinated SMAD7 was then immunoprecipitated with anti-Flag M2 beads and incubated with purified OTUD1-wt/CA protein as indicated time. Lysates were analyzed by IB with anti-HA-Ub, anti-SMAD7, and anti-OTUD1 antibodies **f**. Flag-SMAD7 and His-Ub were transfected into HEK293T cells. Subsequently, poly-ubiquitinated SMAD7 from the cell lysate was pulled down by Nickle beads and incubated with purified OTUD1-wt/CA for 60 mins. Lysates were analyzed by IB with anti-SMAD7 and anti-OTUD1 antibodies, asterisk indicates free SMAD7 **g**. His-SMAD7 was first incubated with E1, E2 (UbcH6), GST-RNF12, and ubiquitin in vitro for 3 h at 37 °C. The poly-ubiquitinated His-SMAD7 was immunoprecipitated by anti-SMAD7 antibody then incubated with purified OTUD1-wt/CA for 60 mins. The assay mixtures were analyzed by anti-Ub, anti-SMAD7 and anti-OTUD1 antibodies (**h**). *$p < 0.05$, **$p < 0.01$, and ***$p < 0.001$ (two-tailed Student's t-test **a**, **b** or two-way analysis of variance (ANOVA) **b**)

or presence of the proteasome inhibitor MG132 (Fig. 5a). RNF12 promoted poly-ubiquitination of SMAD7 in control cells or OTUD1-CA-expressing cells, but not in OTUD1-wt-expressing cells (Fig. 5b). Depletion of endogenous OTUD1 promoted SMAD7 poly-ubiquitination both at the basal and at the RNF12-induced level (Fig. 5c).

Using CRISPR/Cas9 technology, we generated OTUD1 knock-out HEK293T cells. When endogenous SMAD7 was precipitated in these cells, we observed an accumulation of SMAD7 poly-ubiquitination in OTUD1$^{-/-}$ cells compared with OTUD1$^{+/+}$ cells. This effect was reversed when OTUD1 expression was restored (Fig. 5d). To investigate whether OTUD1 affects SMAD7 turnover, we measured SMAD7 protein stability. Pulse-chase labeling experiments showed that the endogenous SMAD7 displayed a severely impaired half-life in OTUD1-depleted cells compared to wild-type cells (Fig. 5e). Ectopic expression of OTUD1-wt, but not OTUD1-CA, prolonged the half-life of

endogenous SMAD7 (Supplementary Fig. 4c). In line with this, the turnover rate of SMAD7, as measured by cycloheximide (CHX) treatment, was reduced by OTUD1 but enhanced in OTUD1-depleted cells (Fig. 5f; Supplementary Fig. 4d). Taken together, endogenous OTUD1 contributes to SMAD7 stability through its deubiquitination.

**OTUD1 cleaves K33-poly-ubiquitin chain on SMAD7 Lysine 220.** By mass spectrometry, we found that SMAD7 was ubiquitinated on Lysine 220 (Fig. 6a; Supplementary Data 4), in close proximity to the unique poly-proline-tyrosine "PPPY" motif that mediates the recruitment SMURF2 to SMAD7[33] (Fig. 6b). Conjugation of ubiquitin on this site might therefore affect the ability of SMAD7 to interact with SMURF2. To determine the type of ubiquitin linkage on this residue, we co-transfected SMAD7 and OTUD1 with individual ubiquitin expression constructs that can

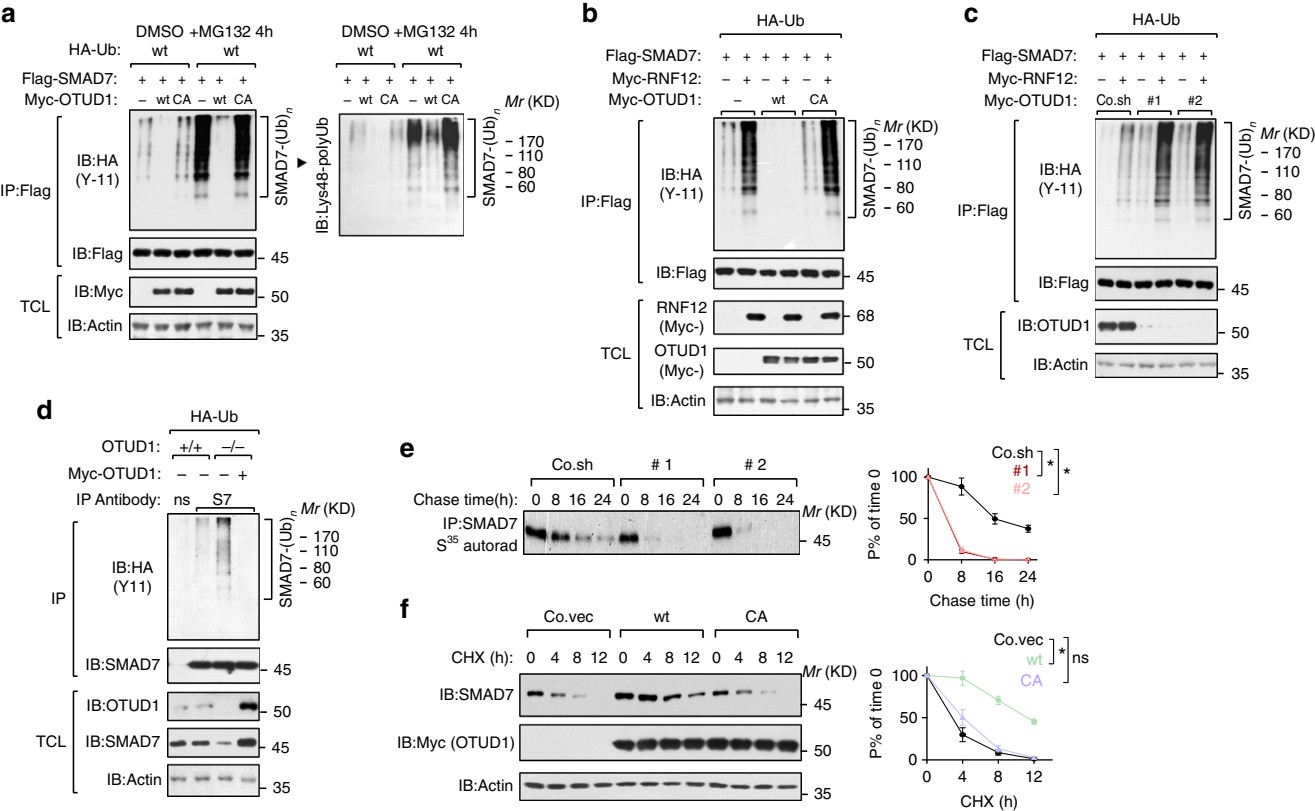

**Fig. 5** OTUD1 deubiquitinates SMAD7 in vivo and sustains SMAD7 stability. **a** IB of total cell lysate (TCL) and immunoprecipitates derived from HEK293T cells transfected with HA-Ub, Flag-SMAD7, Myc-OTUD1-wt/CA and treated with control DMSO or MG132 (5 μM for 4 h) as indicated. **b, c** IB of TCL and immunoprecipitates derived from HEK293T stably expressing HA-Ub and transfected with Flag-SMAD7, Myc-RNF12, and Myc-OTUD1-wt/CA (**b**) or depleted for OTUD1 with shRNA (#1 and #2) **c** as indicated. Poly-ubiquitinated SMAD7 was immunoprecipitated with anti-Flag M2 beads and analyzed by IB with anti-HA-Ub antibody. **d** IB of TCL and immunoprecipitates derived from $OTUD1^{+/+}$ and $OTUD1^{-/-}$ HEK293T cells stably expressing HA-Ub and restored with or without Myc-OTUD1 expression plasmid. Endogenous poly-ubiquitinated SMAD7 was immunoprecipitated with anti-SMAD7 (S7) antibody and immunoblotted with anti-HA-Ub antibodies. **e** [$^{35}$S]-methionine labeling and pulse-chase studies of SMAD7 in control (Co.sh) and OTUD1-depleted PC3 (#1 and #2). The amount of immunoprecipitated labeled protein after the chase was expressed as the percentage of that at the beginning of the chase (time 0) and shown in the right panel. Results are shown as means ± SD of two independent sets of experiments in duplicate. **f** IB of lysates derived from PC3 cells stably expressing empty vector (Co.vec), Myc-OTUD1-wt or OTUD1-CA and treated with cycloheximide (CHX, 20 μg/ml) at the indicated time points. Actin was analyzed as an internal loading control. Quantification of the band intensities was shown in the right panel. Band intensity was normalized to the $t = 0$ controls. Results are shown as means ± SD of three independent sets of experiments. *$p < 0.05$ (two-tailed Student's $t$-test **e**, **f**)

only form ubiquitin linkages at one lysine (for example, K27- indicates every lysine except K27 is changed to Arginine). As shown (Fig. 6c), SMAD7 was primarily conjugated with K27-, K33-, and K48- ubiquitin chains and all of those linkage could be antagonized by OTUD1. Mutation of Lysine 220 to Arginine (K220R) on SMAD7 completely abolished its K33-poly- ubiquitination but had no obvious effect on K27- or K48-poly- ubiquitination (Fig. 6d), suggesting that K220 is the major site specifically responsible for K33-poly-ubiquitination on SMAD7.

The NMR structure of the SMAD7 PY motif region in complex with the SMURF2 WW3 domain revealed that the residues C-terminal to the PY motif (PY-tail), including D217, are involved in direct WW3 domain binding[33]. K220 locates in a flexible region right after the PY motif in SMAD7. Structure modeling suggested that a covalent conjugated ubiquitin molecule on K220 is likely to occupy the interaction space between the PY-tail and the β1-strand and β1–β2 loop of the WW3 domain (Fig. 6e, middle and lower panels), thereby interfering with SMURF2 binding. Considering the compact conformation of K33-linked poly-ubiquitin chain[34], a poly-ubiquitin chain has higher possibility to obstruct the SMURF2 interaction. This suggest that

K33-linked poly-ubiquitination on K220 might block SMAD7–SMURF2 binding. Thus, OTUD1-mediated removal of K33-linked poly-ubiquitination on K220 is likely needed for the SMAD7 PY motif to recruit SMURF2. We next set to confirm this hypothesis with experiments. In OTUD1-depleted cells, we precipitated endogenous SMAD7 and observed accumulated K33-linked poly-ubiquitination (Fig. 6f). Moreover, OTUD1-loss accumulated K33-linked poly-ubiquitination was only observed with SMAD7-wt protein and not with SMAD7 K220R (Fig. 6g). These results corroborate that OTUD1 is a DUB that cleaves K33-linked poly-ubiquitination of SMAD7 K220. We then compared SMAD7-wt and SMAD7 K220R and indeed found SMURF2 to strongly associate with SMAD7 K220R mutant (Fig. 6h). In line with this, knockdown of OTUD1, which results in increased K33-poly-ubiquitin conjugation on SMAD7 K220, reduced the ability of SMURF2 to interact with SMAD7 (Fig. 6i). This effect of the SMAD7 K220 mutation was reflected in a direct readout of TGF-β signaling as the K220R mutant is more potent in suppressing the TGF-β/SMAD-induced CAGA-Luc reporter (Fig. 6j).

Upon activation of TGF-β signaling, SMAD7 binds TβRI as part of a negative feedback response[10,11], and recruits SMURF2 to

promote TβRI turnover at the plasma membrane[12]. Given our results that OTUD1 enables SMAD7/SMURF2 function, we examined whether OTUD1 misexpression can regulate TβRI levels at the cell surface, the location where intracellular signaling is initiated. When OTUD1 was depleted, biotin-labeled cell surface TβRI receptor displayed a prolonged half-life (Supplementary Fig. 5a). In line with this finding, ectopic expression of

OTUD1-wt, but not the OTUD1-CA mutation, led to accelerated TβRI degradation (Supplementary Fig. 5b). Moreover, SMAD7 K220R showed stronger capacity than SMAD7-wt to shorten the half-life of TβRI at the cell surface (Fig. 6k). Together, these findings indicate that OTUD1 is a DUB that removes K33-poly-ubiquitination on SMAD7 K220, which promotes SMAD7's inhibitory role on antagonizing TβRI (Fig. 6l).

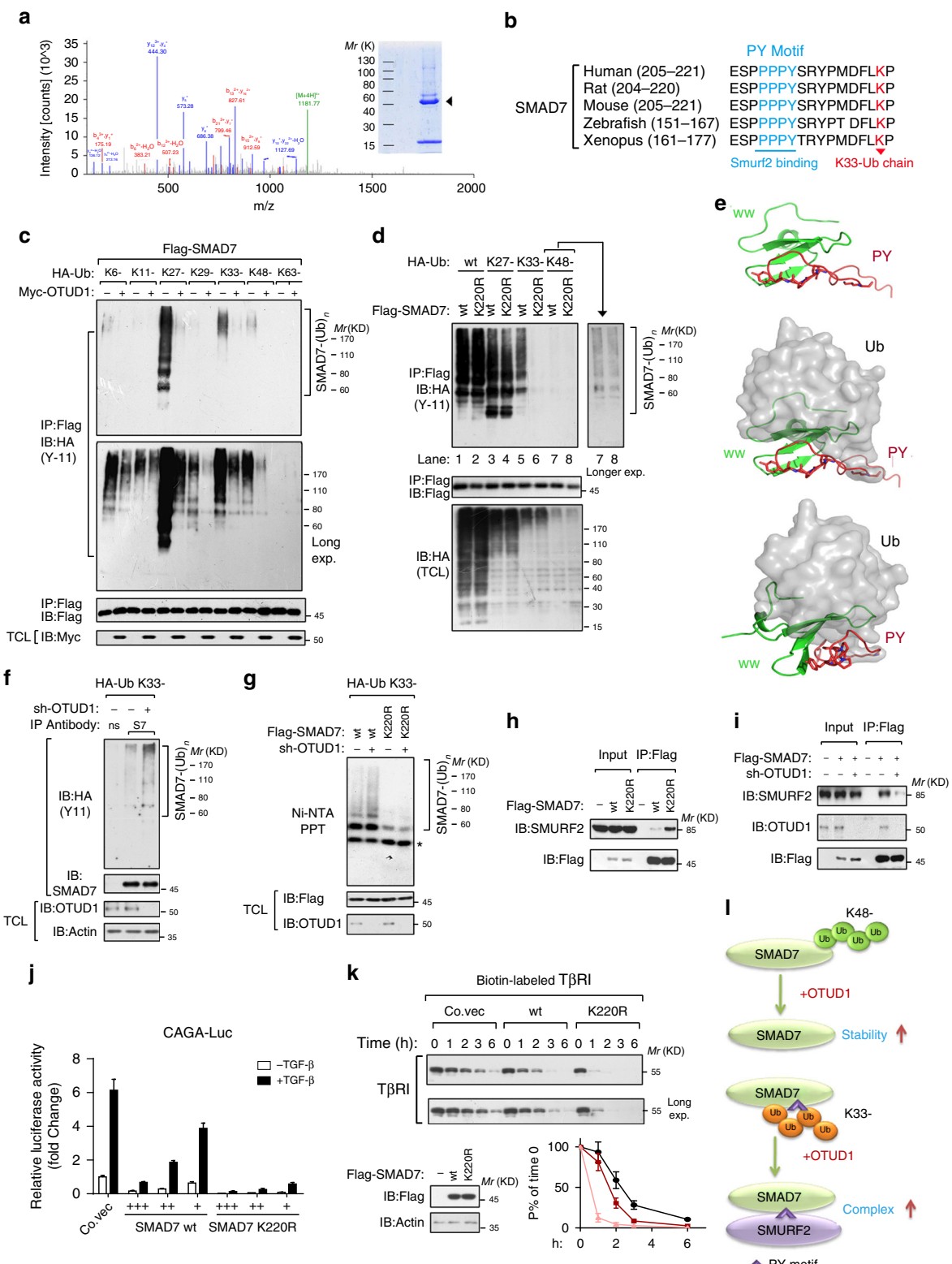

**OTUD1 inhibits metastasis in transplantable mouse models**. To verify our MDA-MB-231 xenograft results we investigated the effect of OTUD1 on lung metastasis using allograft models. Luciferase-labeled control or OTUD1-depleted 4T07 cells were injected intravenously into nude mice and subjected to bioluminescent imaging (BLI). OTUD1-depleted cells exhibited significantly increased lung metastasis abilities around two weeks (Fig. 7a; Supplementary Fig. 6a). Continued BLI monitoring revealed a further enhancement of metastatic outgrowth in the lungs of animals injected with OTUD1-depleted cells (Fig. 7a), and histological analyses indicated a significant increase in the number of metastatic lesions and also in the average surface areas produced by OTUD1-depleted cells when compared with the control cells (Fig. 7b, c; Supplementary Fig. 6b). Taken together, these analyses show that loss of OTUD1 strongly promotes breast cancer lung metastasis.

Considering the importance of the immune system in lung metastasis, we extended our analysis to an immunocompetent mouse model of lung metastasis. We manipulated OTUD1 expression with either ectopic expression or knockdown strategies in the 4T1 murine breast cancer cell line (Supplementary Fig. 6c) and examined their ability to influence metastasis in vivo. Stable cells that ectopically expressed OTUD1-wt or OTUD1-CA were introduced into nude mice through tail vein injection; mice were subjected to BLI. Compared to the control cells, lung metastasis was reduced in OTUD1-wt-expressing cells, but not in the OTUD1-CA-expressing cells (Fig. 7e–g). It was clear that both the number and the average lesion surface of the lung metastasis nodules are significantly reduced in OTUD1-wt group (Fig. 7f–i). Interestingly, we also observed an increase in SMAD7 expression in OTUD1-wt when compared with control or OTUD1-CA tumors (Fig. 7i). To mimic the pathological condition of tumorigenesis and metastasis, we then injected 4T1 cells under the nipple of BALB/c mice, which allows us to examine both primary tumor growth and spontaneous lung metastasis. Here OTUD1-depleted 4T1 cells showed increased spontaneous lung metastasis without affecting primary tumor growth (Fig. 7j–m; Supplementary Fig. 6d). qRT-PCR analysis indicated upregulation of cancer stem cell markers and EMT genes in OTUD1-depleted 4T1 lung-metastatic nodules when compared with control group (Supplementary Fig. 6e), again suggesting that OTUD1 functions to oppose cancer stemness and EMT-related gene-expression programs. Therefore, our studies suggest that OTUD1 inhibits lung metastasis in transplantable mouse models.

**Loss of OTUD1 gene copies correlates with poor prognosis**. To determine the clinical relevance of the above findings in advanced human cancers, we first analyzed the TCGA database and found that the *OTUD1* gene is lost or even deleted in many tumor types as revealed by GISTIC analysis[35], the total percentage is more than 50% in glioblastoma, melanoma and lung cancer (Fig. 8a; Supplementary Table 2). Expression of OTUD1 showed positive correlation with its gene copy number in breast cancer. Meanwhile, we observed that the very rare breast cancer patients with gain or amplification of OTUD1 have co-occurrence with p53 expression and mutual exclusivity with expression of PIK3CA and AKT1 (Fig. 8b; Supplementary Fig. 6f). Combined with our previous analysis, these observations strongly favor the notion that OTUD1 is a metastasis suppressor.

In addition, we investigated whether OTUD1 transcription is downregulated in cancer. Similar as the BRCA1 and BRCA2 tumor suppressors, expression of *OTUD1* was significantly suppressed by wild-type HRAS in p53 null MCF10A cells, but not in the control MCF10A cells (p53-wt), indicating that gain of oncogenic RAS together with loss of p53 tumor suppressor could lead to decreased OTUD1 function (GSE81593) (Fig. 8c). Consistent with the notion that activation of HRAS inhibits OTUD1 expression in p53 mutant cells, we found that upon ectopic expression of constitutively active HRASV12, *OTUD1* is downregulated in p53 mutant MDA-MB-231 (both the early passage and the bone-metastatic (BM) cell lines) but not in p53 wt MCF7 cells (Fig. 8d), TGF-β and HRAS collaborate to promote EMT, invasion and cancer stem traits in breast cancer cells[36]. In line with this, depletion of OTUD1 enhanced the effect of HRAS in MCF10A cells (Fig. 8e; Supplementary Data 3).

From surgical resections, we collected 100 patient-derived samples of breast cancer for tissue microarray analysis. The immunohistochemical (IHC) analysis of OTUD1 and SMAD7 levels revealed a statistically significant positive correlation (Fig. 8f–h). In a large public clinical microarray database of human breast tumors, we found a trend towards good prognosis for OTUD1-high patients (Fig. 8i). Low OTUD1 correlates with poor survival in patients with lymph node invasion/metastasis. But for the patients without lymph node signal, OTUD1 failed to distinguish prognosis (Fig. 8j), suggesting a more significant role of OTUD1 in antagonizing metastatic tumors. These findings are in support of our hypothesis that OTUD1 shuts off breast cancer metastasis by deubiquitinating SMAD7.

## Discussion

Metastasis is a process in which cancer cells relocalize to another organ. To reach distant organs, circulating tumor cells (CTCs) that have intravasated from the primary tumor must overcome many obstacles through mechanisms that are not well understood[37]. Recent

**Fig. 6** OTUD1 cleaves K33-poly-ubiquitin chain on SMAD7 Lysine 220. **a** K220 ubiquitination of SMAD7 identified by mass spectrometry. Right: Comassie staining of Flag-SMAD7 precipitations. **b** Sequence alignment of the PY motif and identified K220 ubiquitination site in SMAD7 orthologs of different species. **c** Immunoblot (IB) of total cell lysate (TCL) and immunoprecipitates derived from Flag-SMAD7 overexpressing HEK293T cells transfected with K6-, K11-, K27-, K29-, K33-, K48-, and K63-linked HA-Ub constructs and Myc-OTUD1 as indicated. **d** IB of TCL and immunoprecipitation derived from HEK293T cells transfected with wt, K27-, K33-, K48-linked HA-Ub, together with Flag-SMAD7-wt or K220R mutant as indicated. **e** Upper panel: the model of WW domain of E3-ubiquitin ligase Smurf2 binding to PY loop (residue 203-220) of SMAD7. Middle and lower panels: the model of mono-ubiquitin (PDB ID: 2XK5) conjugation to PY loop (residue 203-220) at residue Lys220. Both WW domain (green) and PY loop (red) were shown in ribbon, ubiquitin was shown in gray surface, and the key residues involved in domain interaction are drawn in stick representations. **f** IB of TCL and anti-SMAD7 immunoprecipitants derived from K33-linked HA-Ub expressing HEK293T cells depleted of OTUD1. **g** IB of TCL and Nickle-pull down precipitants derived from K33-linked His-Ub expressing HEK293T cells transfected with Flag-SMAD7-wt/ K220R and depleted of OTUD1 as indicated. **h** IB of input and anti-Flag immunoprecipitates derived from HEK293T cells transfected with Flag-SMAD7-wt or K220R mutant as indicated. **i** IB of input and anti-Flag immunoprecipitates derived from HEK293T cells transfected with Flag-SMAD7 and depleted of OTUD1 as indicated. **j** CAGA$_{12}$-Luc transcriptional response of HEK293T cells transfected with empty vector (Co.vec), SMAD7-wt or K220R at different doses as indicated and treated with TGF-β (0.5 ng/ml) overnight. **k** IB of biotinylated cell surface TβRI in HeLa cells stably transfected with empty vector (Co. vec), SMAD7-wt or K220R mutant expression plasmids and treated with TGF-β (5 ng/ml) for indicated time points. Quantification of the band intensities is shown in the lower panel. Band intensity was normalized to the $t = 0$ controls. Results are shown as means ± SD of three independent sets of experiments. **l** Working model of OTUD1-mediated SMAD7 deubiquitination

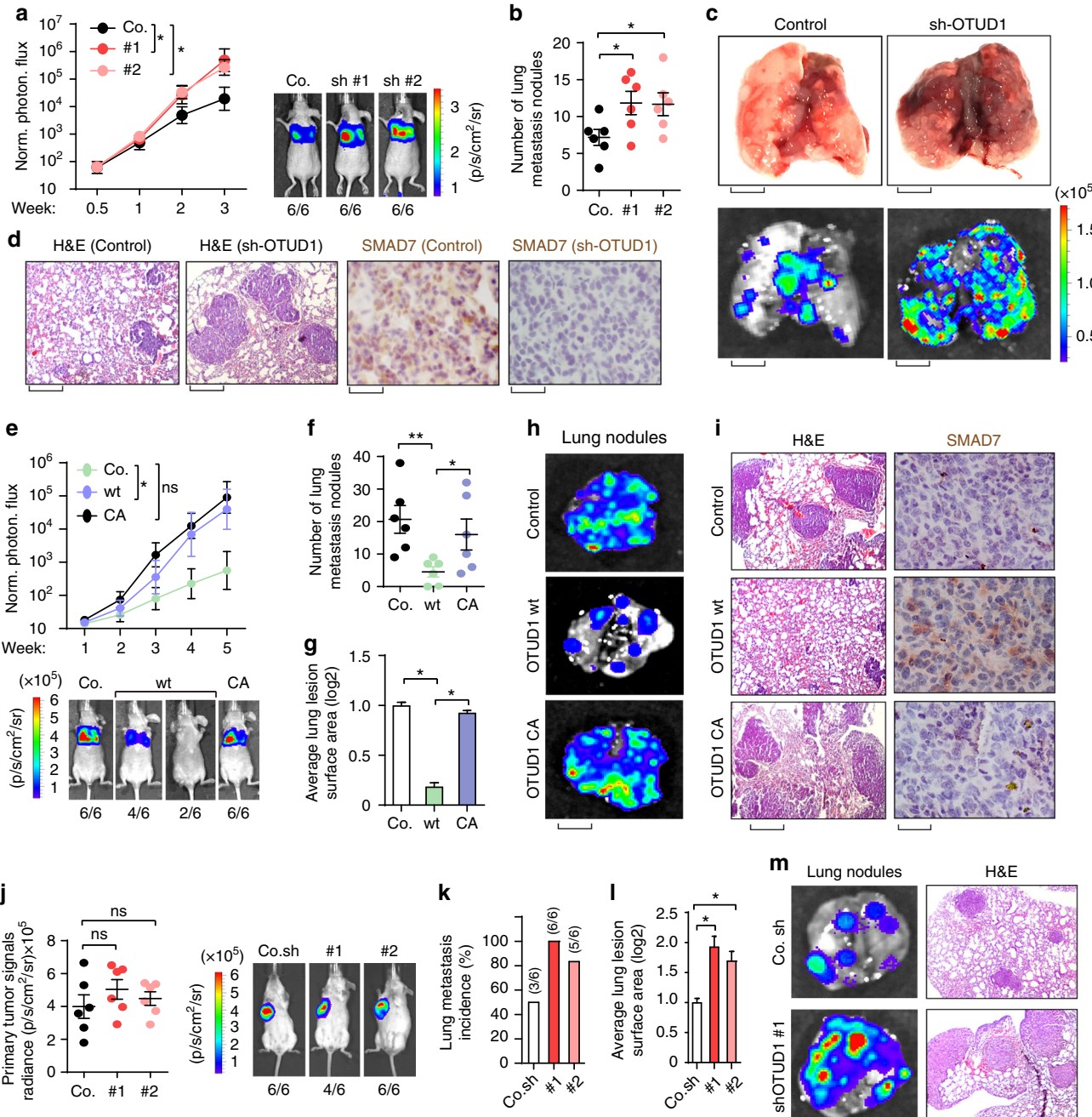

**Fig. 7** OTUD1 inhibits lung metastasis in transplantable mouse models of breast cancer metastasis. **a–d** Control (Co.) or OTUD1-silenced (sh#1 and sh#2) 4T07 cells were tail vein-injected into nude mice. Lung metastasis was measured by BLI. Normalized photon flux at the indicated times left (±SD) and representative images (right) are shown; p-values; Student's t-test (**a**). Lung metastasis nodules in each group were calculated (±SD) and the representative images of bright view (**c**, upper panel; scale bar, 2 mm) and bioluminescent view (**c**, lower panel; scale bar, 2 mm) are shown. Representative HE and IHC staining pictures of lung sections of control and one shRNA-depleted OTUD1 group (**d**), scale bars, 1 mm and 20 μm for lung haematoxylin and eosin (H&E) and SMAD7 immunohistochemistry images, respectively. **e–i** Control (Co.) 4T1 cells or cells stably expressing OTUD1-wt or CA were tail vein-injected into nude mice. Lung metastasis was measured by BLI. Normalized photon flux (±SD) at the indicated times are shown in the upper panel and representative images in the lower; p values; Student's t-test **e**. Lung metastasis nodules (±SD) **f** and average lung lesion surface area (±SD) **g** (arbitrary units based on pixel quantification from digital images) in each group were calculated and the representative images of bioluminescent view (**h**, scale bar, 2 mm) are shown. Representative HE and IHC staining pictures of lung sections of every experimental group (**i**, Scale bars, 1 mm and 20 μm for lung haematoxylin and eosin (H&E) and SMAD7 immunohistochemistry images, respectively) are shown. **j–m** Control (Co.) or OTUD1-depleted 4T1 cells were orthotropic injected in syngeneic BALB/c mice. Primary tumor formation in mice after orthotropic injection of each group at day 21 after implantation. Normalized BLI signals (±SD) (Left) and representative BLI signals of primary tumor. Number of lung metastasis nodules (**k**) and average lung lesion surface area (±SD) (**l**) (arbitrary units based on pixel quantification from digital images). Representative bioluminescent view of isolated lung (**m**, left, scale bar, 2 mm) and HE staining of lung section of each group (**m**, right, scale bar, 20 μm). *p < 0.05 and **p < 0.01 (two-tailed Student's t-test **a–l**)

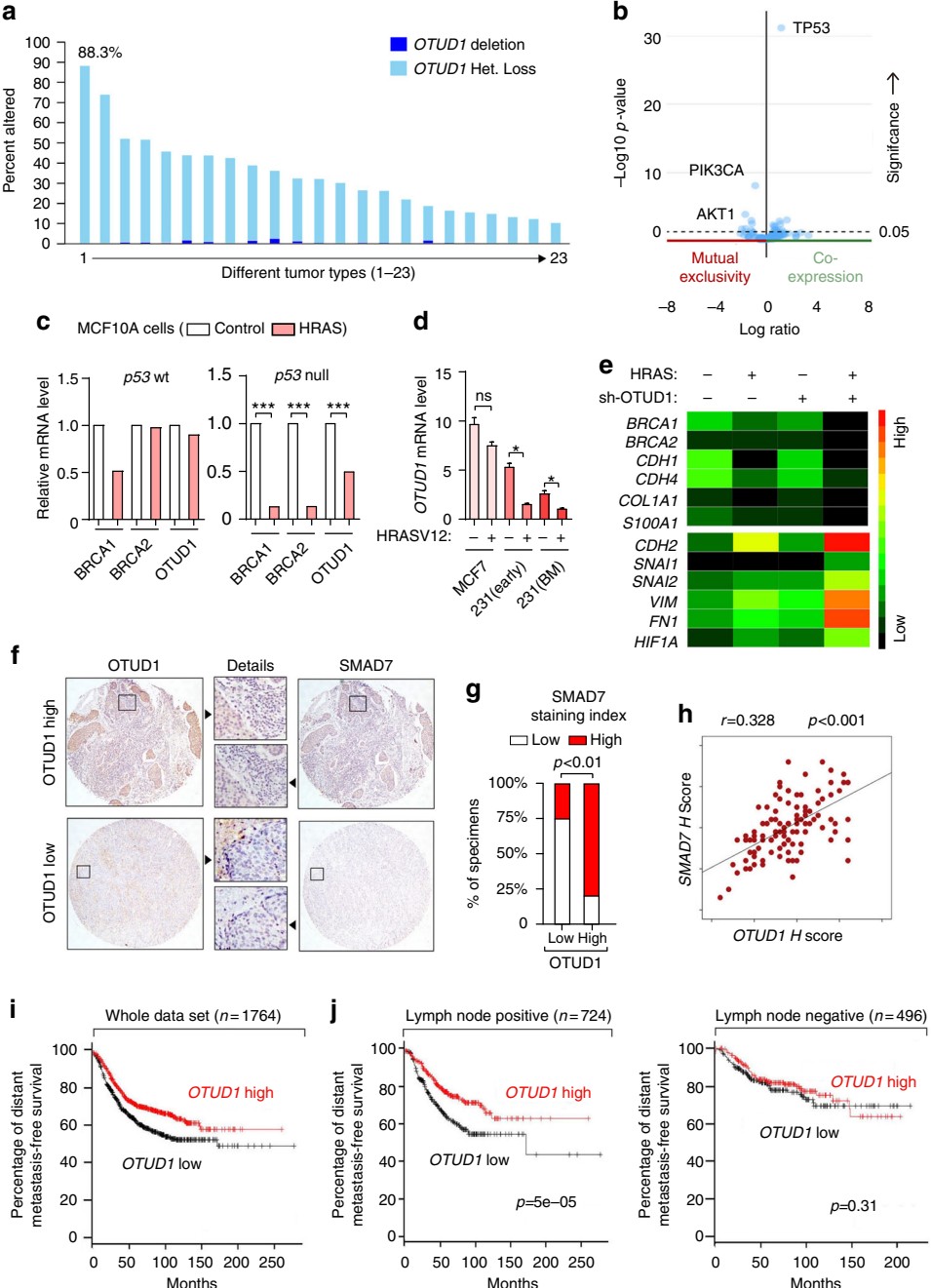

**Fig. 8** OTUD1 correlates with SMAD7 in human specimens and the loss of OTUD1 gene copies in human cancer correlates with poor prognosis. **a** Graph showing *OTUD1* gene copy loss or deletion in different tumor types. Data were obtained from the TCGA database[65, 66]. Detailed values are listed in Supplementary Table 2. **b** Volcano plot showing that the rare breast cancer patients with gain or amplification of OTUD1 have co-occurrence with p53 expression and mutual exclusivity with expression of PIK3CA and AKT1. Data were obtained from the TCGA database[65, 66]. **c** Analysis of *BRCA1*, *BRCA2*, and *OTUD1* mRNA level in p53 wt- and p53 null- MCF10A cells expressed with or without wild-type HRAS (GSE81593 dataset). **d** qPCR analysis of *OTUD1* mRNA level in MCF7, MDA-MB-231 (early passage), MDA-MB-231 bone-metastatic (BM) cells that ectopically express HRASV12 or control vector. Values and error bars represent the mean ± SD of triplicates and are representative of at least two independent experiments. **e** Heat map of mRNA level of listed genes in MCF10A cells that stably express control or HRAS with depletion of OTUD1 (sh-OTUD1), as indicated. **f–h** Immunohistochemistry analysis of OTUD1 and SMAD7 in breast cancer tissue microarrays; representative pictures of each antibody staining are shown (**f**). Percentage of specimens displaying low or high SMAD7 expression compared to the expression levels of OTUD1 (**g**). Scatterplot showing the positive correlation between OTUD1 and SMAD7 expression in patients; Pearson's coefficient tests were performed to assess statistical significance **h**. **i** Kaplan–Meier curves showing that the metastasis-free survival of individuals was positively correlated with OTUD1 expression. **j** Kaplan–Meier curves showing that the metastasis-free survival of individuals was positively correlated with OTUD1 expression in lymph node positive patient (Left) and has no correlation with OTUD1 in lymph node negative patients (Right). *$p < 0.05$ and ***$p < 0.001$ (two-tailed Student's *t*-test (**d**, **g**), Pearson's coefficient tests (**h**) or Log-rank test (**i**, **j**))

findings have just started to define the extracellular cues, phenotypic properties, hosting niches, and signaling pathways that support the survival, self-renewal, dormancy, and reactivation of cancer cells that initiate metastasis: metastatic stem cells (MSC)[38]. By dissecting the biology of this multistep process, vulnerabilities are being exposed that could be exploited to prevent metastasis. To enable search of factors that favor metastatic outgrowth, we developed an in vivo screening system in mice to select for genetic determinants that promote metastasis. Our loss of function screen shares some similarity to other screens that have recently been reported using shRNA- or a CRISPR/Cas9-based approaches to identify modulators of metastasis using intravenous injection or subqutaneous transplant delivery of tumor cells, respectively[39,40]. In this study, we screened a shRNA library that covers human DUB family members that inhibit metastasis and use an intracardial administration route of tumor cells. From our screen, we have identified that loss of OTUD1 enables breast cancer cells to undergo EMT and gain cancer stem cell traits thereby driving metastatic spread to distant organs, such as bone and lung. OTUD1 was subsequently identified as a potent negative regulator of the TGF-β signaling pathway, a strong inducer of EMT and cancer cell stemness.

DUBs antagonize the action of E3-ubiquitin ligases, of which the latter mediate the Ub modification of certain target proteins. DUBs remove Ub post-translational modifications of proteins, thereby regulating virtually all Ub-dependent processes[41]. Both DUBs and E3 ligases are emerging classes of enzymatic drug targets[42,43]. Although K48-linked Ub chains generally serve as a proteasomal degradation signal[41], K63-linked chains are non-degradative and, for example, activate protein kinase cascades[44]. Lys11 linkages constitute an alternative degradation signal used during cell-cycle progression[45]. Met1-linked chains cooperate with K63 linkages in NF-κB signaling[46]. But the cellular roles of the remaining types (K6, K27, K29, and K33) are elusive[47].

The deubiquitinase OTUD1 was found to enable the antagonistic role of SMAD7 on TGF-β signaling. SMAD7 recruits SMURF2 to the activated TβRI at the cell surface to promote its ubiquitination and degradation. Moreover, SMAD7 itself is ubiquitinated by RNF12 and ARKADIA for proteasomal degradation. OTUD1 displayed two layers of regulation and specificity on SMAD7 poly-ubiquitination: OTUD1 removes K48-chain thereby inhibiting SMAD7 degradation. Moreover, OTUD1 cleaves the K33-chain on SMAD7 K220 residue thereby exposing the SMAD7 PY-tail binding pocket for interaction with the WW domain of SMURF2. By doing so, OTUD1 enables SMAD7 function thereby antagonizing TGF-β/SMAD signaling. Our finding is consistent with the previous finding that OTUD1 is capable of cleaving multiple types of ubiquitin linkage[48]. Of note, the K27-chain strongly conjugates on SMAD7 and is also cleaved by OTUD1, suggesting a "third" layer of SMAD7 regulation by OTUD1 that requires further investigation. By generating SMAD7-deficient clone of the early passage MDA-MB-231 cells, we found that the anti-metastatic role of OTUD1 is dependent on endogenous SMAD7 expression. Although OTUD1 could not show significant inhibition of metastasis in SMAD7-deficient cells, our results could not exclude the possibility that OTUD1 might also target other substrates. Although OTUD1-wt can completely cleave K48-linked poly-ubiquitin chains from SMAD7 in DMSO-treated cells, this effect is not observed in the MG132-treated cells, suggesting that there might be certain DUBs present at the proteasome that have a role in deubiquitinating SMAD7.

As mentioned the K33-linked ubiquitin linkage is an atypical one. Here we show by structural modeling and follow-up biochemical and cell biological experiments that K33-linked poly-ubiquitination of SMAD7 might shield the PY interaction motif, and thereby interferes with protein–protein interaction. We validated this hypothesis by showing that mutation of the

SMAD7 K220 site could completely abolish SMAD7 K33-linked poly-ubiquitination (Fig. 6d) and sharply increased SMAD7's binding affinity with SMURF2 (Fig. 6h). As far as we know, such a mechanism of action has not been reported before for K33-linked ubiquitination. In contrast to our study, others have reported that K33-linked ubiquitination can actually mediate the recruitment of regulatory proteins[49–51], or that such chains compete with degradative K48-linked ubiquitination[52]. Of note, monoubiquitination of SMAD4 interferes with the interaction with phosphorylated SMAD2 and thereby inhibits TGF-β signaling[53].

Another aspect that deserves discussion is the broad and frequent loss of OTUD1 expression in human cancers. We observed that some breast cancers express aberrantly low concentrations of OTUD1 as a result of OTUD1 gene loss or even gene deletion and that the loss of OTUD1 expression in patient correlates with a poor prognosis. Gain of RAS activity and loss of p53 expression are often the characteristics of advanced tumors. Gain of HRAS expression and the loss of p53 signaling have been shown to collaborate with each other in promoting the loss of OTUD1 expression in breast cancer cells. Importantly, OTUD1 gene loss is also present in many other types of cancers, such as glioblastoma and lung cancers, indicating that the tumor suppressing role of OTUD1 may not be limited to breast cancer. OTUD1 is targeted for gene loss in around 50% of lung tumors (Fig. 8a). Patients with lung carcinoma and low OTUD1 gene expression have shorter lifetimes than those with lung cancer but high OTUD1 expression (Supplementary Fig. 6g)[54]. Moreover, OTUD1 correlated with the SMAD7 expression in our breast cancer tissue microarray analysis, further corroborating that OTUD1 has an essential role in the control of SMAD7 stability and activity. Moreover, in both xenograft and orthotopic mouse cancer models, loss of OTUD1 expression is closely associated with breast cancer metastasis.

Combined, the in vitro and in vivo evidence in this study elucidates critical functions of OTUD1 in the termination of TGF-β/SMAD-induced metastatic activation, indicating that OTUD1 restricts the EMT and cancer stem cell traits during metastasis. Moreover, as an in vivo deubiquitinase of SMAD7, OTUD1 represents a novel and critical mechanism that controls intrinsic SMAD7 activity and thereby TGF-β/SMAD signaling.

## Methods

**Cell culture**. HEK293T, HeLa, HaCaT, MCF7, 4T1, 4T07, MDA-MB-231 cell lines were originally from ATCC and cultured in Dulbecco's modified Eagle's medium (DMEM) supplemented with 10% FBS and 100 U/ml penicillin–streptomycin. MCF10A (MI) and MCF10A-RAS (MII) cell lines were obtained from Dr. Fred Miller (Barbara Ann Karmanos Cancer Institute, Detroit, USA) and cultured in DMEM/F12 supplemented with 5% horse serum, 20 ng/ml epidermal growth factor (EGF), 0.5 μg/ml hydrocortisone, 100 ng/ml cholera toxin, and 100 U/ml penicillin–streptomycin. All the cells are free of mycoplasma contamination.

**Lentiviral transduction and generation of stable cell lines**. Lentiviruses were produced by transfecting HEK293T cells with shRNA-targeting plasmids and the helper plasmids pCMV-VSVG, pMDLg-RRE (gag/pol), and pRSV-REV. The cell supernatants were harvested 48 h after transfection and were either used to infect cells or stored at −80 °C.

To obtain stable cell lines, cells were infected at low confluence (20%) for 24 h with lentiviral supernatants diluted 1:1 with normal culture medium in the presence of 5 ng/ml of polybrene (Sigma). Forty-eight hours after infection, cells were placed under puromycin selection for one week and then passaged before use. Puromycin was used at 2 μg/ml to maintain MDA-MB-231 cells, MCF10A-RAS, 4T1 and 4T07 cells. Lentiviral shRNAs were obtained from Sigma (MISSION® shRNA). Typically, five shRNAs were identified and tested, and the most effective two shRNAs were used for the experiments. We used TRCN0000350761 (#1) and TRCN0000350762 (#2) for human OTUD1 knockdown and designed two primer pairs for mouse OTUD1 knockdown as indicated below:

Primer1-Forward: CCGGCAGATGCTGAATGT
GAATATACTCGAGTATATTCACATTCA GCATCTGTTTTTG; Primer1-

Reverse: AATTCAAAAACAGATG CTGAATGTGAATATACTC GAGTATATTCACATTCA GCATCTG;

Primer2-Forward: CCGGGCTCAGCAAT GGACACTATGACT CGAGTCATAGTGTCCATTG CTGAGCTTTTTG;

Primer2-Reverse: AATTCAAAAAGC TCAGCAATGGACACTATGACTCG AGTCATAGTGTCCATTGCTGAGC.

**Transcription reporter assay**. HEK293T cells were seeded in 24-well plates and transfected with the indicated plasmids using calcium phosphate. Luciferase activity was measured with a PerkinElmer luminometer. The internal transfection control *renilla*, was used to normalize luciferase activity. Each experiment was performed in triplicate, and the data represent the mean ± SD of three independent experiments.

**Immunoprecipitation and immunoblotting**. Cells were lysed with 1 ml lysis buffer (20 mM Tris-HCl pH 7.4, 2 mM EDTA, 25 mM NaF, and 1% Triton X-100) containing protease inhibitors (Sigma) for 10 min at 4 °C. After centrifugation at $12 \times 10^3 \times g$ for 15 min, the protein concentrations were measured, and equal amounts of lysate were used for immunoprecipitation. Immunoprecipitation was performed with anti-FLAG M2 beads (Sigma, A2220) for 1 h at 4 °C or with different antibodies and protein A-Sepharose (GE Healthcare BioSciences AB) for 3 h at 4 °C. Thereafter, the precipitants were washed three times with washing buffer (50 mM Tris-HCl pH 8.0, 150 mM NaCl, 1% Nonidet P-40, 0.5% sodium-deoxycholate, and 0.1% SDS), and the immune complexes were eluted with sample buffer containing 1% SDS for 5 min at 95 °C. The immunoprecipitated proteins were then separated by SDS-PAGE. Immunoblot (IB) analysis was performed with specific antibodies and secondary anti-mouse or anti-rabbit antibodies conjugated to horseradish peroxidase (Amersham Biosciences). Visualization was achieved with chemiluminescence.

For the analysis of cell surface receptors, the proteins at the cell surface were biotinylated for 40 min at 4 °C and then incubated at 37 °C for the indicated times. The biotinylated cell surface receptors were precipitated with streptavidin beads and analyzed by immunoblotting. All the uncropped and unmodified scans of western blot data were shown as Supplementary Figs. 7–9 in the Supplementary Information.

**In vivo screening and mouse metastasis assay**. Up to 371 shRNA lentivirus (and also control shRNA) were produced in HEK293T cells. Virus were then introduced into MDA-MB-231-GFP/Luc cells individually. Forty-eight hours after infection, cells were placed under puromycin selection for another 3 days. Control and 371 shRNA stable cells were then harvested and mixed into single-cell suspensions ($1 \times 10^5 /100$ μl PBS) then were inoculated into the left heart ventricle of 5-weeks-old female BALB/c nude mice ($n = 30$) according to the method described by Arguello et al[55]. Development of metastasis was monitored weekly by bioluminescent reporter imaging. Early-onset metastasis nodules were isolated and genomic DNA was purified for sequencing. According to standard protocol (TIANamp Genomic DNA Kit, Cat. #DP304-03) the metastatic cells were harvested in lysis buffer GA containing RNaseA and then incubated with Proteinase K. After removing the RNA and protein, the genomic DNA was purified by TIANamp Spin Column CB3 followed by extensive wash with buffer PW. Details can be found in: http://tiangen.com/asset/imsupload/up0875005001348194139.pdf. No randomization and blinding was used for mice experiments.

**OTUD1 knockout by CRISPR/Cas9**. sgRNA design and cloning was performed according to the Feng Zhang lab general cloning protocols[56,57]. OTUD1 sgRNAs oligos and SMAD7 sgRNA oligos were designed based on the target site sequence (20 bp) and are flanked on the 3′ end by a 3 bp NGG PAM sequence. Using the Cas9 target design tools (http://www.genome-engineering.org), we designed two sgRNAs for each target:

OTUD1 sg1 forward: 5′-CACCGAGGAGGAGCAG CCGGGGACG-3′;
OTUD1 sg1 reverse: 5′-AAACCGTCCCCGGCTG CTCCTCCTC-3′;
OTUD1 sg2 forward: 5′-CACCGGTCCACAGGACGATGTGCAG-3′;
OTUD1 sg2 reverse: 5′-AAACCTGCACATCGTCCTGTGGACC-3′;
SMAD7 sg forward: 5′-CACCCCAAACGATCTGCGCTCGTCG-3′; SMAD7 sg reverse: 5′-AAACGACGAGCGCAGATCGTTTGGC-3′;

The sgRNAs were cloned into the lentiCRISPRv2 vector (Addgene). For lentivirus production, cloned lentiCRISPRv2 plasmids were co-transfected into HEK293T cells with the packaging plasmids pVSVg (AddGene 8454) and psPAX2 (AddGene 12260). Lentivirus were harvested and MDA-MB-231 cells were infected with two sgRNA mixtures for OTUD1. Forty-eight hours after infection, cells were placed under puromycin selection for 1 week and the single-cell-derived clones were picked, expanded and knockout of OTUD1 was verified by IB analysis.

**GSEA**. We used GSEA v2.0 to perform GSEA on various functional and/or characteristic gene signatures. Gene sets were obtained from the MSigDB database v3.0 (September 2010 release). Statistical significance was assessed by comparing the enrichment score to enrichment results generated from 1,000 random permutations of the gene set to obtain $p$ values (nominal $p$ value). Data from NKI 295, a well-annotated human breast cancer database, were analyzed for the enriched gene signature, and *OTUD1*-high (OTUD1 ≥ 0.1, $n = 54$) versus *OTUD1*-low (OTUD1 ≤ −0.1, $n = 64$) samples were compared.

**Primers and reagents**. The DNA primer sequences that were used to detect target gene expression by qRT-PCR are listed in Supplementary Data 5. Doxycycline-inducible pCW-Myc-OTUD1-wt/CA, pLv-puro-Myc-OTUD1-wt/CA and pCDNA3.1-Flag-OTUD1-wt/CA constructs were cloned and verified by DNA sequencing. MG132 was purchased from Sigma. Rhodamine phalloidin was purchased from Molecular Probe, Invitrogen. The antibodies used for immunoprecipitation, immunoblotting, and immunofluorescence were the following: N-cadherin 1:5,000 (IB; Clone 32, BD Biosciences), fibronectin 1:1,000 (IB; Sigma), vimentin 1:1,000 (IB; # 5741 Cell Signaling) and E-cadherin 1:10,000 (IB; BD 610181), 1:500 (IF; BD 610181), c-Myc 1:1,000 (IB; a-14, sc-789, Santa Cruz Biotechnology), c-Myc 1:1,000 (IB; Y-11, sc-805, Santa Cruz Biotechnology), HA 1:1,000 (IB; Y-11, sc-805, Santa Cruz Biotechnology), HA 1:10,000 (IB; 12CA5, in-house), Flag 1:10,000 (IB; M2, Sigma), Ub 1:1,000 (IB; P4D1, Santa Cruz Biotechnology), OTUD1 1:1,000 (IB; Abcam 122481), SMAD7 1:1,000(IB; sc-7004;Santa Cruz, this antibody was validated to recognize only SMAD7, not SMAD6), RNF12 1:1,000 (ab57230, Abcam and M0), SMAD4 1:1,000 (IB; B8, Santa Cruz), SMAD2/3 1:2,500 (IB); 1:500 (IP; 610842 BD), phospho-SMAD2 1:5,000 (IB; # 3101, Cell Signaling); Actin 1:10,000 (A5441, Sigma).

**Real-time RT-PCR (qRT-PCR)**. Total RNAs were prepared using the NucleoSpin® RNA II kit (BIOKÉ, Netherlands). A total of 1 μg of RNA was reverse-transcribed using the RevertAid™ First Strand cDNA Synthesis Kit (Fermentas). Real-time PCR was conducted with SYBR Green (Applied Bioscience) using a StepOne Plus real-time PCR system (Applied Bioscience). All target gene-expression levels were normalized to *GAPDH*.

**Immunohistochemical staining and evaluation**. Primary antibodies to OTUD1 (1:50; Abcam 122481), SMAD7 (Specificity were tested to read endogenous SMAD7 at 1:200; sc-7004, Santa Cruz Biotechnology) were used for immunohistochemical staining of formalin-fixed paraffin-embedded microarrays of breast cancer tissues (obtained from US Biomax (BC081116c)), according to previously described staining protocols[58]. The quantification of staining was expressed as an H score. The H score was determined by the formula 3 × the percentage of strongly staining cells + 2 × the percentage of moderately staining cells + the percentage of weakly staining cells, yielding a range of 0 to 300. Scoring was performed blindly by three independent investigators.

**Nickel pulldown assay**. Cells were transfected with His-Ub plasmid and followed by nickel bead purification and immunoblot analysis[59]. For His-SMAD7 purification, the washed cell pellet was resuspended in lysis buffer (6 M guanidine-HCl, 0.1 M $Na_2HPO_4/NaH_2PO_4$, and 10 mM imidazole). After sonication and a freeze-thaw step, the supernatant of the cell lysate was incubated with Talon beads (BD Biosciences) in the presence of 20 mM imidazole. Beads were washed four times with lysis buffer lacking imidazole. Purified proteins were eluted in lysis buffer containing 200 mM imidazole.

**Protein purification**. SMAD7 protein and RNF12 proteins were generated in *E. coli* as His-tagged and GST fusion protein, respectively, as previously described[13]. For purification of OTUD1-wt/CA protein from mammalian cells, OTUD1-wt/CA expression plasmids were transfected in HEK293T cells and immunoprecipitated overnight with α-Flag-M2 resin (Sigma), followed by elution with Flag peptide (Sigma, 1 mg/ml in 50 mM HEPES (pH 7.5), 100 mM NaCl, 0.1% NP40, 5% glycerol).

**In vitro binding assay**. For in vitro protein–protein interactions, purified and/or recombinant proteins were diluted in binding buffer (25 mM HEPES [pH 7.5], 100 mM KCl, 2 mM $MgCl_2$, 0.1% NP40, 5% glycerol), immunoprecipitated with anti-OTUD1 or anti-SMAD7 for 3 h and protein A-Sepharose beads for 1 h, and washed three times with the same buffer. Lysates were then analyzed by IB for coprecipitating proteins.

**In vitro deubiquitination of SMAD7**. In vitro ubiquitin assays were carried out by incubating ubiquitin-activating enzyme E1 (Ube1 40 nM), Ubiquitin (8 μM), UbcH6 (0.7 μM), 500 ng His-SMAD7, 500 ng GST-RNF12, and 500 ng OTUD1-wt/CA proteins at 37 °C for 3 h in reaction buffer, 50 mM Tris-HCl (pH 7.5), 5 mM $MgCl_2$, 1 mM DTT, 2 mM ATP with proteases inhibitor. 9 μl of the 10 μl reaction system were diluted with RIPA buffer followed by immunoprecipitation and anti-Ubiquitin immunoblotting. 1 μl of the 10 μl reaction system was loaded as input control. In Fig. 4f, Poly-HA-ubiquitinated Flag-SMAD7 substrate was prepared by transiently cotransfecting HEK293T cells with Flag-SMAD7 and HA-ubiquitin. Thirty-six h later, cells were treated with MG132 (5 μM) for 4 h and then the Poly-HA-ubiquitinated Flag-SMAD7 was purified in denatured conditions. To assay the ability of OTUD1 protein to deubiquitinate SMAD7 in vitro, purified OTUD1 protein was incubated with purified Poly-HA-ubiquitinated Flag-SMAD7 at 37 °C for indicated time. Reaction was terminated by SDS sample buffer followed by a 2 min heat denaturation at 95 °C. Reaction products were detected by IB analysis. For

the experiment shown in Fig. 4g, Poly-His-ubiquitinated Flag-SMAD7 was purified by nickel beads pulldown.

**In vivo ubiquitination assay**. As previously described[60], HEK293T cells were transfected with indicated constructs and treated with or without MG132 (5 μM) for 4 h before harvesting. 48 h after transfection, cells were washed with PBS and lysed in two pellet volumes of RIPA buffer (20 mM NAP, pH 7.4, 150 mM NaCl, 1% Triton, 0.5% Sodium-deoxycholate, and 1% SDS) supplemented with protease inhibitors and 10 mM N-Ethylmaleimide (NEM). Lysates were sonicated, boiled at 95 °C for 5 min, diluted with RIPA buffer containing 0.1% SDS, then centrifuged at 4 °C ($16 \times 10^3 \times g$ for 15 min). The supernatant was incubated with specific antibody and protein A-Sepharose for 3 h at 4 °C. After extensive washing, bound proteins were eluted with 2×SDS sample buffer and separated on a SDS-PAGE, followed by IB analysis.

**Tumor sphere assays**. As described[61], single-cell suspensions of MCF10A (RAS) cells ($1 \times 103$ cells/ml) were plated on ultra-low attachment plates and cultured in phenol red-free DMEM/F12 (Gibco Paisley, UK; 21041) containing B27 supplement (no vitamin A; Invitrogen, Paisley, UK; 12587) and rEGF (20 ng/ml; Sigma Aldrich Poole, UK; E-9644). Tumor spheres were visualized using a phase contrast microscope, photographed, and counted.

**3D matrigel assays**. Cells were cultured in growth-factor-reduced reconstituted basement membrane (Matrigel, Corning) as described previously[62]. Images were taken with an inverted microscope.

**Mice metastasis models**. Mice experiments were approved by the Committee for Animal Welfare in Zhejiang University. Mice were purchased from animal husbandry center of the Shanghai Institute of Cell Biology, Academia Sinica, Shanghai, China. For the intracardial injection assay, five-weeks-old female BALB/c nude were anesthetized with isofluorane and single-cell suspension of MDA-MB-231/luc cells ($1 \times 10^5$/100 μl PBS) were inoculated into the left heart ventricle according to the method described by Arguello et al[55]. Bioluminescent imaging was used to verify successful injection and to monitor the metastatic outgrowth. After 6 weeks, all mice were killed and metastatic lesions were confirmed by histological analysis. For the doxycycline-inducible metastatic reactivation, MDA-MB-231 cells were infected with pCW-Myc-OTUD1-wt or CA mutant and intracardial inoculated into 5-weeks-old female BALB/c nude mice. Doxycycline was administered to mice at day 21 after injection of the cancer cells, through the diet (625 mg/kg of food) as well as by intraperitoneal injection (25 mg/kg of body weight; three times a week). For tail vein injection, single-cell suspensions of 4T1-Luc or 4T07-Luc cells ($1 \times 10^5$/100 μl PBS) cells were injected into the tail vein of five-weeks-old nude mice or female syngeneic BALB/c mice. Development of metastasis was monitored weekly by bioluminescent reporter imaging. After 6 weeks, all mice were killed and lung metastases were dissected. Mouse nipple implantation was based on a previously published method[63]. Female BALB/c mice were anesthetized and used for this assay. Overall, $1 \times 10^5$ 4T1-Luc cells were injected through the nipple area into the mammary fat pad. At 21 days after injection, luciferin was injected and the primary tumors were analyzed, then the mice were euthanized and analyzed for acquisition of secondary tumor(s). All primary/metastatic tumors were detected by BLI with the IVIS 100 (Caliper Life Sciences, Hopkinton, MA, USA). The BLI signal intensity was quantified as the sum of photons within a region of interest given as the total flux (photons/s). No randomization and blinding was used for mice experiments.

**Pulse-chase**. As previously described[13], cells were plated in 6-well plates, starved for 3 h in Met/Cys-free medium and pulsed for 40 min with 200 μCi/ml of $^{35}$S labeled Met/Cys. After two washes, the cells were chased in medium supplemented with cold (unlabeled) Met and Cys (100 μg/ml) before harvesting. Endogenous proteins were collected from the extracts by immunoprecipitation, which were thereafter resolved by SDS-PAGE and visualized by autoradiography. Each experiment was performed in duplicate.

**Mass spectrometry**. As previously described[64], SDS-PAGE gels were minimally stained with Coomassie brilliant blue, cut into six molecular weight ranges based on heavy chain IgG bands, and digested with trypsin. Immunocomplexes were identified on a Thermo Fisher LTQ (majority) or Velos-Orbitrap mass spectrometer. Spectral data were then searched against the human protein RefSeq database in BioWorks or the Proteome Discoverer Suites using either SeQuest (for LTQ data) or Mascot (Orbitrap data) software. The IP/MS results were transferred into a FileMaker-based relational database generated in-house, where protein identification numbers (protein GIs) were converted to Gene ID identifiers according to the NCBI "gene accession" table.

**Structural modeling**. The NMR structure of WW domain binding to PY loop (residue 203–217) was obtained from Protein Data Bank (PDB ID: 2DJY). Three more residues of PY loop including Lys220 were manually added, and the ubiquitin (PDB ID: 2XK5) was modeled using WinCoot. The structure refinement was done using CCP4. This was performed by the refinement and structure idealization options in the software to idealize the protein geometry. All the figures are made by PyMOL.

**Statistical analyses**. Statistical analyses were performed with a two-tailed unpaired $t$-test or as indicated in the legends. $p$ value is indicated by asterisks in the figures: $^*p < 0.05$; $^{**}p < 0.01$; $^{***}p < 0.001$. Differences at $p = 0.05$ and lower were considered significant. No randomization or blinding was used.

**Data availability**. The authors declare that all data supporting the findings of this study are available within the article and its Supplementary Information files.

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

## Acknowledgements

We are grateful to Martijn Rabelink, Erik Meulmeester, Alfred Vertegaal, Joost Schimmel, and David Baker for providing reagents and for valuable discussions. We thank A.G. Jochemsen for early passage MDA-MB-231 cells. We thank Midory Thorikay for expert technical assistance. This work was supported by a special program from Ministry of Science and Technology of China (2016YFA0502500 to L.Z.), the Chinese National Natural Science Funds (31571460 to F.Z., 31471315 and 31671457 to L.Z.), PCSIRT (IRT1075 to X.G.), Jiangsu National Science Foundation (BK20150354 to F.Z.), the key R&D projects of Zhejiang province (2015C03045) and Zhejiang outstanding youth fund (LR14C070001 to L.Z. and LR15C060001 to J.J.).

## Author contributions

Z.Z., F.X., Y.F., K.J., L.S., W.S., H.v.D., P.F, B.Y. and X.Z. performed the experiments; H.Z. and S.Y. performed the structure analysis, Z.Z., H.v.D., X.Y., M.Z., J.Jia, J.Jin, and C.D. created the figures; F.Z., P.t.D. and L.Z. wrote the paper.

## Additional information

**Competing interests:** The authors declare no competing financial interests.

