## [Peer Review File · Nature Communications]

Reviewer #1 (Remarks to the Author):

In the present study, the authors screened deubiquitinating enzymes for suppressors of breast cancer metastasis and identified OTUD1. They confirmed this finding using several different in vivo metastasis models and database analyses. They further explored the underlying mechanism. They concluded that suppression of metastasis is attributed to its negative regulation of TGF- β signaling by deubiquitinating Smad7 (This reviewer has some concerns on this argument. See below.) These findings would contribute to the tumor biology field.

Importantly they also unveiled the role of K33-linked ubiquitination in negative regulation of Smad7 function. This is a novel and interesting finding as well. It would attract attention of researchers of the TGF- β signaling field and also ubiquitin field.

Experiments were technically sound, well controlled, and mostly support their conclusions. However, this reviewer feels some gaps in their arguments.

- 1) Loss of OTUD1 resulted in induction of mesenchymal and stem-cell-like properties of breast cancer cells. Is it because of enhancement of TGF- β signaling? The authors examined the effect of a TGF- β receptor kinase inhibitor on mesenchymal marker expressions in Figure 3G, but not stem-cell-like properties. So far, effects of TGF- β on breast cancer stem cells have been controversial; some support positive roles (Mani et al., Cell 133, 704-715, 2008) while others support negative roles (Tang et al., Cancer Res. 67, 8643-8652, 2007). Therefore, this question should be experimentally addressed before moving onto Figure 3.
- 2) OTUD1 inhibits TGF- β signaling and the authors concluded that the suppression is attributed to enhancement of Smad7 function. This reviewer feels that this might not be the whole story. Does OTUD1 inhibit TGF- β signaling even after knockdown of Smad7? The authors extensively explored how OTUD1 affects stability and function of Smad7, but before these experiments, they should show that Smad7 is a major target of OTUD1 when it suppresses TGF- β signaling.
- 3) Similarly, does OTUD1 fail to behave as a metastasis suppressor even in the absence of Smad7? OTUD1 suppresses metastasis and also deubiquitinates Smad7. However, it remains unclear if there is a causal relationship. This is an important point because it would directly support the title of this paper. In addition, Smad7 is known to have functions independent of suppressing TGF- β family signaling. Even if ideas of authors on point 1 and/or 2 do not prove to be right, point 3 is still worth trying.

Minor points

- 4) Figure 1B: What happened to 6/30 mice? Some explanation would be helpful.
- 5) Figure 4A: In this figure, Smad proteins appear to be C-terminally tagged, which is usually avoided because it may affect C-terminal phosphorylation of R-Smads. Is Figure 4A necessary? Binding specificity of Smad proteins to OTUD1 is clearly shown in Figure 4D.
- 6) Figure 5D: Show Smad7 expression in TCL. It is important to compare endogenous Smad7 expression in OTUD1 wt and knockout cells.
- 7) Figure 7: Scale bars are not explained at all.
- 8) Figure 8C and D: In Figure 8C, the authors stated that H-Ras downregulated OTUD1 in p53-null MCF10A cells, but not in p53-wt MCF10A cells. In Figure 8D, H-Ras down-regulated OTUD1 in MII and MDA-MB-231 cells. MII cells are p53-wt cells. Why was OTUD1 down-regulated? This reviewer asks some explanation. In line 372, only a result using MDA-MB-231 cells was described. The result of MII should be included. In addition, MII cells are H-Ras-transduced cells and MDA-MB-231 cells are KRAS mutant cells. This reviewer wonders why H-Ras was still effective, especially in MII cells. MII cells express very high level of H-Ras protein.
- 9) Line 84 and 110, the term "OUT domain" is not used in other papers. They use "OTU domain".
- 10) Line 112, MDA-MB-231 (BM) cells are not described in Materials and Methods. Some reference(s) should be cited.
- 11) Line 153-155, Figures 2D-E are data on MDA-MB-231 cells, according to the figure legend.
- 12) Line 229, 231 and overall, both ubiquitination and ubiquitylation were used.
- 13) Line 289, structural modeling should be mentioned in Materials and Methods.
- 14) Line 368, is H-Ras used here wt or mutated?
- 15) Line 372, for readers' convenience, the authors should mention that MDA-MB-231 cells are p53-mutant cells and MCF7 cells are p53-wt cells.
- 16) Line 419, ARKADEA >> ARKADIA
- 17) Lin2 420, OUTD1 >> OTUD1
- 18) Line 465 and others, 100 U/ml-1 may be 100 U/ml or 100U ml-1.
- 19) Line 875-877, PDB IDs (WW-PY and Ub) should be described in the legend.
- 20) Line 933, in the legend to Figure 8D, MCF10A was described but in Figure 8D, MII was used.

Reviewer #2 (Remarks to the Author):

To the authors:

The authors use cell-based assays, biochemistry, animal studies, and human cancer databases to demonstrate that OTUD1 deubiquitinates a specific substrate, SMAD7 and rescues it from ub-mediated degradation. Loss of OTUD1 results in increased SMAD7 degradation, resulting in increased TGF β signaling and a stemness/metastasis phenotype. The studies are largely persuasive. Can the authors address these comments?

1) if you go back to the original shRNA assay in MDA-MB-231 cells and knockdown SMAD7, does this result in a phenocopy of knocking down OTUD1 (i.e., get increased metastasis)?

2) In the experiment in Figure 2C, does lentiviral mediated expression of wt SMAD7 also, like wt OTUD1, reduce the number of tumor-like colonies?

3) A prediction is that an OTUD1 transgenic mouse (like a SMAD7 transgenic mouse) would suppress metastasis. Another prediction is that an OTUD1(-/-) mouse, like a SMAD7(-/-) mouse (if viable), would have increased stemness and increased metastasis. Has this been shown?

4) if you were to re-express an shRNA-resistant OTUD1 cDNA back into cells derived from the OTUD1-depleted cells, would this re-expression of OTUD1 (but not CA mutant) reduce the metastatic potential of the cells?

5) In the Discussion, it is important to point out that OTUD1 probably has many substrates.

Reviewer #4 (Remarks to the Author):

Author Comments

The manuscript by Xie et. al. describes a series of experiments to establish that the DUB OTUD1 acts as a metastasis suppressor. OTUD1 stabilizes SMAD7 to inhibit TGF-beta pathway. OTUD1 also cleaves K33-linked poly-ubiquitin chains at lysine 220 of SMAD7 to expose its PY motif, leading to SMURF binding and regulation of TbetaRI turnover. The experiments were performed in a technically sound manner (except the structural modeling), and the data is convincing. However, the differential expression/activity of DUBs in cancers and metastasis are well known. The mechanism of such DUBs is to stabilize tumor/metastasis suppressors or promote interactions that maintain homeostasis. OTUD1 appears to stabilize SMAD7 by a similar mechanism. Moreover, the mechanism of OTUD1 promoted SMAD7/SMURF binding is not strong (see below). This paper is suited for a more specialized journal.

Major points

- 1) The structural modeling based mechanism of poly-ubiquitination at K220 inhibiting the interaction at PY motif is not convincing. The K220 is in a flexible region, and the conjugated C-terminal tail of Ub is also flexible. Unless there is a specific interaction between the PY motif and the conjugated Ub, the conjugated poly-ubiquitin chain will be dynamic and may not occlude SMAD7/SMURF interaction. The authors need to show a specific interaction between the PY motif and Ub to validate the mechanism described in Figure 6E.
- 2) The author claim that this study reports a new mechanism of action by K33-linked poly-ubiquitin chains. It is unclear what is the relevance of K33 polyubiquitin chains in the suggested mechanism? By their modelling studies, it appears that mono-ubiquitin is sufficient to inhibit SMAD7/SMURF interaction. Then why has the system evolved to generate poly-ubiquitin chains of a specific linkage at K220?
- 3) The K48-linked polyubiquitination of SMAD7 appears to be significantly less than K27-linked polyubiquitination (Fig 6C). Hence, rather than stabilizing SMAD7 from proteasomal degradation, a more profound role of OTUD1 could be altering the function of SMAD7-(K27-linked)-polyUb by its DUB activity. K27-linked poly-ubiquitin chains have been reported to promote protein-protein interactions, tumor growth, etc. Unfortunately, the function of SMAD7-(K27-linked)-polyUb has been ignored here.

Minor Points

- 1) In the Figure 4G legend, "arrow" is mentioned as a marker for free SMAD7. In the figure, it is shown by "asterisk".

- 2) In Fig 4H, in-vitro ubiquitination and de-ubiquitination were done at the same time. This does not clarify if OTUD1 interacts with RNF12 and inhibits polyubiquitination. Instead, the authors should first ubiquitinate SMAD7, deplete ATP and then observed de-ubiquitination.
- 3) The authors often say OTUD1 “inhibits” SMAD7 poly-ubiquitination, whereas it deubiquitinates SMAD7. Inhibiting ubiquitination is a completely different mechanism, which is generally not a DUB activity.
- 4) Figure 5A right lane. While wt-OTUD1 can completely cleave K48-linked –SMAD7 in the DMSO treated cells, it does not in the MG132 treated cells. Do the DUBS present at the proteasome have a role in de-ubiquitinaing SMAD7?
- 5) The authors use both “ubiquitination” and “ubiquitylation”. It is preferable to stick to any one of these terms.
- 6) Correct spelling “uibiquitination” in line 259.
- 7) Line 277, introduce a blank between “withSMURF2”.

Manuscript number: NCOMMS-17-00755-T

Rebuttal to comments by the reviewers

We are grateful to the constructive and valuable comments raised by the reviewers. In responding to them our manuscript has strongly improved.

Reviewer #1 (Remarks to the Author):

In the present study, the authors screened deubiquitinating enzymes for suppressors of breast cancer metastasis and identified OTUD1. They confirmed this finding using several different in vivo metastasis models and database analyses. They further explored the underlying mechanism. They concluded that suppression of metastasis is attributed to its negative regulation of TGF- β signaling by deubiquitinating Smad7 (This reviewer has some concerns on this argument. See below.) These findings would contribute to the tumor biology field. Importantly they also unveiled the role of K33-linked ubiquitination in negative regulation of Smad7 function. This is a novel and interesting finding as well. It would attract attention of researchers of the TGF- β signaling field and also ubiquitin field.

Response: Many thanks for comments.

Experiments were technically sound, well controlled, and mostly support their conclusions. However, this reviewer feels some gaps in their arguments.

Response: Many thanks for comments; please see our point to point answers below.

1) Loss of OTUD1 resulted in induction of mesenchymal and stem-cell-like properties of breast cancer cells. Is it because of enhancement of TGF- β signaling? The authors examined the effect of a TGF- β receptor kinase inhibitor on mesenchymal marker expressions in Figure 3G, but not stem-cell-like properties. So far, effects of TGF- β on breast cancer stem cells have been controversial; some support positive roles (Mani et al., Cell 133, 704-715, 2008) while others support negative roles (Tang et al., Cancer Res. 67, 8643-8652, 2007). Therefore, this question should be experimentally addressed before moving onto Figure 3.

Response: In response to reviewer, we now show in Supplementary Figure S3A and S3B that the stem-like properties of breast cancer cells, as measured by tumor organoids and mammosphere sphere assays, were promoted by TGF- β ligand stimulation and were inhibited by the treatment of T β RI kinase inhibitor SB431542. These results are described on Page 8, the last paragraph before we started moving onto Figure 3.

2) OTUD1 inhibits TGF- β signaling and the authors concluded that the suppression is attributed to enhancement of Smad7 function. This reviewer feels that this might not be the whole story. Does OTUD1 inhibit TGF- β signaling even after knockdown of Smad7? The authors extensively explored how OTUD1 affects stability and function

of Smad7, but before these experiments, they should show that Smad7 is a major target of OTUD1 when it suppresses TGF- β signaling.

Response: In response to reviewer, we generated SMAD7 knockout cells by using CRISPR/Cas9 technology (Supplementary Figure S4A). We found that the ectopic expression of OTUD1 suppressed TGF- β signaling in control cells but demonstrated no significant effect in SMAD7 deficient cells (Figure 4A, left panel); The knock down of OTUD1 promoted TGF- β signaling in control cell but not in SMAD7 deficient cells (Figure 4A, right panel). These results indicate that the SMAD7 is a major target of OTUD1 in suppressing TGF- β signaling.

3) Similarly, does OTUD1 fail to behave as a metastasis suppressor even in the absence of Smad7? OTUD1 suppresses metastasis and also deubiquitinates Smad7. However, it remains unclear if there is a causal relationship. This is an important point because it would directly support the title of this paper. In addition, Smad7 is known to have functions independent of suppressing TGF- β family signaling. Even if ideas of authors on point 1 and/or 2 do not prove to be right, point 3 is still worth trying.

Response: In response to this reviewer and also being asked by the second reviewer, we generated a SMAD7 knockout cell line of the early passage MDA-MB-231 cells by guide RNA of the CRISPR/Cas9 technology (Supplementary Figure S4A). Mice injected with SMAD7 knockout MDA-MB-231 clone developed relative rapid and stronger metastases and had significantly shorter metastases-free survival periods (Figure 4B). Moreover, ectopic expression of OTUD1 significantly inhibited metastasis in control cells but did not show significant inhibitory effect in the SMAD7 knockout cells. This demonstrates that the anti-metastatic activity of OTUD1 is largely mediated by regulating SMAD7 function. Even though, our results do not exclude possibilities that OTUD1 might have other substrates.

Minor points

4) Figure 1B: What happened to 6/30 mice? Some explanation would be helpful.

Response: The 6/30 mice developed weak micrometastasis. This has now been explained in Page 5 in the revised version of our manuscript. We apologize for not being clear.

5) Figure 4A: In this figure, Smad proteins appear to be C-terminally tagged, which is usually avoided because it may affect C-terminal phosphorylation of R-Smads. Is Figure 4A necessary? Binding specificity of Smad proteins to OTUD1 is clearly shown in Figure 4D.

Response: We have deleted the old Fig 4A according to reviewer's suggestion.

6) Figure 5D: Show Smad7 expression in TCL. It is important to compare endogenous Smad7 expression in OTUD1 wt and knockout cells.

Response: Blotting of the endogenous SMAD7 is now shown in Fig 5D in the revised version.

7) Figure 7: Scale bars are not explained at all.

Response: All the scale bars have been explained. Many thanks for pointing this out for us. We apologize for this missing information in the figure legends.

8) Figure 8C and D: In Figure 8C, the authors stated that H-Ras downregulated OTUD1 in p53-null MCF10A cells, but not in p53-wt MCF10A cells. In Figure 8D, H-Ras down-regulated OTUD1 in MII and MDA-MB-231 cells. MII cells are p53-wt cells. Why was OTUD1 down-regulated? This reviewer asks some explanation. In line 372, only a result using MDA-MB-231 cells was described. The result of MII should be included. In addition, MII cells are H-Ras-transduced cells and MDA-MB-231 cells are KRAS mutant cells. This reviewer wonders why H-Ras was still effective, especially in MII cells. MII cells express very high level of H-Ras protein.

Response: Figure 8C showed results from GSE81593 dataset in which the wild-type HRAS was used; we made this clear in the figure legend. In the Figure 8D, we used the constitutive active H-RASV12 mutant. We also made this clear in the Figure and manuscript.

Although the MII cells are stable HRAS transduced cells, we could still reach a highly efficient over-expression by using lentiviral mediated infection of HRASV12 (as shown below; Figure A). This suggests that the stable HRAS transduction perhaps already inhibited or bypassed p53 function thus enable the *OTUD1* down-regulation by further highly increased ectopic expression of activated HRASV12.

Figure A. qRT-PCR analysis of *HRAS* mRNA expression from MCF10A-RAS (MII) cells infected with lentivirus encoding control (-) or HRASV12 mutant (+) as indicated.

Even though, we agree with reviewer that MII cells are p53-wt and we do not have clear evidences for our hypothesis mentioned above. Therefore, we removed result of MII cells from Figure 8D and instead showed new result from the early passage MDA-MB-231 cells and the bone metastatic MDA-MB-231 cells, in which ectopic expression of HRASV12 strongly suppressed the expression of *OTUD1*.

9) Line 84 and 110, the term “OUT domain” is not used in other papers. They use “OTU domain”.

Response: We have corrected this mistake, many thanks for pointing it out for us.

10) Line 112, MDA-MB-231 (BM) cells are not described in Materials and Methods. Some reference(s) should be cited.

Response: References were added in the text, many thanks.

11) Line 153-155, Figures 2D-E are data on MDA-MB-231 cells, according to the figure legend.

Response: They are results from MCF10A-RAS cells, we corrected the figure legend. We apologize for this mistake.

12) Line 229, 231 and overall, both ubiquitination and ubiquitylation were used.

Response: All the “ubiquitylation” have been changed into “ubiquitination”, many thanks for pointing this out for us.

13) Line 289, structural modeling should be mentioned in Materials and Methods.

Response: We added structure modeling in the Materials and Methods, many thanks for pointing out this missing point to us.

14) Line 368, is H-Ras used here wt or mutated?

Response: H-Ras that was used is wild-type HRAS. We made this clear in the text.

15) Line 372, for readers' convenience, the authors should mention that MDA-MB-231 cells are p53-mutant cells and MCF7 cells are p53-wt cells.

Response: The statements have been included, many thanks.

16) Line 419, ARKADEA >> ARKADIA

17) Lin2 420, OUTD1 >> OTUD1

18) Line 465 and others, 100 U/ml-1 may be 100 U/ml or 100U ml-1.

Response: All those mistakes were corrected, many thanks.

19) Line 875-877, PDB IDs (WW-PY and Ub) should be described in the legend.

Response: The WW-PY (PDB ID: 2DJY) and Ub (PDB ID: 2XK5) were described in the legend. Thank you for reminding us.

20) Line 933, in the legend to Figure 8D, MCF10A was described but in Figure 8D, MII was used.

Response: It should read MCF10A-RAS (MII) cells. We corrected our mistake.

Reviewer #2 (Remarks to the Author):

To the authors:

The authors use cell-based assays, biochemistry, animal studies, and human cancer databases to demonstrate that OTUD1 deubiquitinates a specific substrate, SMAD7 and rescues it from ub-mediated degradation. Loss of OTUD1 results in increased SMAD7 degradation, resulting in increased TGF β signaling and a stemness/metastasis phenotype. The studies are largely persuasive. Can the authors address these comments?

Response: Many thanks for comments; please see our point to point answers below.

1) if you go back to the original shRNA assay in MDA-MB-231 cells and knockdown SMAD7, does this result in a phenocopy of knocking down OTUD1 (i.e., get increased metastasis)?

Response: In response to this reviewer, we generated SMAD7 knockout cell line of the early passage MDA-MB-231 cells by guide RNA of the CRISPR/Cas9 technology (Supplementary Figure S4A). Mice injected with SMAD7 knockout clone developed relative rapid and stronger metastases and demonstrated significantly shorter metastases-free survival periods (Figure 4B).

2) In the experiment in Figure 2C, does lentiviral mediated expression of wt SMAD7 also, like wt OTUD1, reduce the number of tumor-like colonies?

Response: In response to reviewer, we generated lentiviral construct for SMAD7. Although the lentiviral mediated expression of SMAD7 is not very strong (about 10 fold-increase in SMAD7 mRNA level), it significantly reduced the number of tumor-like colonies (see Figure B).

Figure B. Left panel: Mean number of organoids (\pm SE) per 200 cells from triplicate samples of control and SMAD7 overexpressed MCF10A-RAS cells cultured in 3D Matrigel. Right panel: qRT-PCR analysis of *SMAD7* mRNA expression from sample in the left panel. P values, Student's t test.

3) A prediction is that an OTUD1 transgenic mouse (like a SMAD7 transgenic mouse) would suppress metastasis. Another prediction is that an OTUD1(-/-) mouse, like a SMAD7(-/-) mouse (if viable), would have increased stemness and increased metastasis. Has this been shown?

Response: As far as we know, there is no report of an OTUD1 transgenic mouse or an

OTUD1 knockout mouse. To generate such mice is beyond the scope of the present investigation, which is already dense with data and Figures. In the near future, we would like to generate genetic models with mis-expression of OTUD1 and test this reviewer's hypothesis. We are grateful for the suggestion.

4) if you were to re-express an shRNA-resistant OTUD1 cDNA back into cells derived from the OTUD1-depleted cells, would this re-expression of OTUD1 (but not CA mutant) reduce the metastatic potential of the cells?

Response: This reviewer is asking for a so-called rescue experiment. We used TRCN0000350761 (#1) and TRCN0000350762 (#2) for human OTUD1 knockdown; as shown below (from Sigma website), #2 shRNA is targeting 3'UTR of OTUD1. We therefore tried to rescue #2 shRNA's effect by OTUD1 cDNA, which lacks the 3'UTR sequence.

TRCN0000350762 0.2ml 10⁶TU pLKO.1

Product Details Region:3UTR Mean KnockDown Level: 0.61 Cell Line: 293T/17
 VALIDATED TRC Version: 2 Clone ID:NM_001145373.2-1725s21c1
 Sequence:CCGGTTCAAACTTGCTAGTAGATTTCTCGAGAAATCTACTAGCAAGTTTGAATTTTG

As shown below (Figure C), #2 shRNA-promoted metastatic potential of breast cancer cells could be rescued by ectopic expression of OTUD1 wt, but not CA mutant.

Figure C. Left panel: Mean number of organoids (\pm SE) per 200 cells cultured in 3D Matrigel from triplicate samples of control and OTUD1-silenced (#2) MCF10A-RAS cells infected with lentivirus encoding control vector (Co. vec) or the OTUD1 wt/CA as indicated.

Right panel: Mean number of tumor spheres per 5×10^3 cells from triplicate samples of control and OTUD1-silenced (#2) MCF10A-RAS cells infected with lentivirus encoding control vector (Co. vec) or the OTUD1 wt/CA as indicated. P values, Student's t test.

5) In the Discussion, it is important to point out that OTUD1 probably has many substrates.

Response: In response to reviewer, we mentioned that OTUD1 might also have other substrates in the discussion (In the end of first paragraph, Page 20).

Reviewer #4 (Remarks to the Author):

Author Comments

The manuscript by Xie et. al. describes a series of experiments to establish that the DUB OTUD1 acts as a metastasis suppressor. OTUD1 stabilizes SMAD7 to inhibit TGF-beta pathway. OTUD1 also cleaves K33-linked poly-ubiquitin chains at lysine 220 of SMAD7 to expose its PY motif, leading to SMURF binding and regulation of TbetaRI turnover. The experiments were performed in a technically sound manner (except the structural modeling), and the data is convincing. However, the differential expression/activity of DUBs in cancers and metastasis are well known. The mechanism of such DUBs is to stabilize tumor/metastasis suppressors or promote interactions that maintain homeostasis. OTUD1 appears to stabilize SMAD7 by a similar mechanism. Moreover, the mechanism of OTUD1 promoted SMAD7/SMURF binding is not strong (see below). This paper is suited for a more specialized journal.

Response: Many thanks for comments; please see our point to point answers below.

Major points

1) The structural modeling based mechanism of poly-ubiquitination at K220 inhibiting the interaction at PY motif is not convincing. The K220 is in a flexible region, and the conjugated C-terminal tail of Ub is also flexible. Unless there is a specific interaction between the PY motif and the conjugated Ub, the conjugated poly-ubiquitin chain will be dynamic and may not occlude SMAD7/SMURF interaction. The authors need to show a specific interaction between the PY motif and Ub to validate the mechanism described in Figure 6E.

Response: We agree with the reviewer that K220 locates in a flexible region right after the PY motif, and the conjugated Ub is also flexible. However, the distance is so close that a covalent conjugated ubiquitin molecule on K220 might occupy the interaction space between the PY-tail and the β 1-strand and β 1- β 2 loop of the WW3 domain. We have significantly toned down the statement by suggesting that K33-linked poly-ubiquitination on K220 might block SMAD7-SMURF2 binding (highlighted in Page 14). Indeed, our experimental data shown in Figure 6F-6I support our hypothesis.

2) The author claim that this study reports a new mechanism of action by K33-linked poly-ubiquitin chains. It is unclear what is the relevance of K33 polyubiquitin chains in the suggested mechanism? By their modelling studies, it appears that mono-ubiquitin is sufficient to inhibit SMAD7/SMURF interaction. Then why has the system evolved to generate poly-ubiquitin chains of a specific linkage at K220?

Response: We identified SMAD7 K220 from mass spec study (Figure 6B) as a specific lysine residue for K33-polyubiquitin chain (Figure 6D). Compared to SMAD7-wt, SMAD7 K220R mutant showed enhanced binding affinity with SMURF2 (Figure 6H), thereby reinforcing its capacity to promote the turnover of cell surface T β RI (Figure 6K). K33-linked poly-ubiquitin chain has been suggested to be more compact than other poly-ubiquitin chains. Our structural modelling suggested that a covalent conjugated ubiquitin molecule on K220 likely occupies the interaction space between the PY-tail and

WW3 domain. Therefore, a K33-linked poly-ubiquitin chain on K220 of SMAD7 has higher possibility to interfere with SMURF2 interaction.

3) The K48-linked polyubiquitination of SMAD7 appears to be significantly less than K27-linked polyubiquitination (Fig 6C). Hence, rather than stabilizing SMAD7 from proteasomal degradation, a more profound role of OTUD1 could be altering the function of SMAD7-(K27-linked)-polyUb by its DUB activity. K27-linked poly-ubiquitin chains have been reported to promote protein-protein interactions, tumor growth, etc. Unfortunately, the function of SMAD7-(K27-linked)-polyUb has been ignored here.

Response: K48-linked polyubiquitination mediates SMAD7 degradation through proteasome. Fig. 6C is a comparison in the absence of MG132 treatment, in which most of the K48-linked polyubiquitinated SMAD7 has been targeted for proteasome-mediated degradation. As shown in Fig. 5A, K48-linked polyubiquitination of SMAD7 can be strongly accumulated by MG132 treatment.

As shown in the figure below, if MG132 was added before we harvested cell, K48-linked polyUb chain is comparable with K27-linked polyUb chain on SMAD7.

Figure D. IB of immunoprecipitates derived from HEK293T cells transfected with Flag-SMAD7, HA-Ub K27- or HA-Ub K48- and treated with control DMSO or MG132 (5 μ M for 4 h) as indicated.

As we have mentioned in the discussion (Page 20, end of the first paragraph), we agree with the reviewer that the SMAD7-(K27-linked)-polyubiquitin chain may have important function(s). We would like to investigate its role(s) in the near future, but rather present this in a separate new story.

Minor Points

1) In the Figure 4G legend, “arrow” is mentioned as a marker for free SMAD7. In the figure, it is shown by “asterisk”.

Response: We have corrected the figure legend.

2) In Fig 4H, in-vitro ubiquitination and de-ubiquitination were done at the same time. This does not clarify if OTUD1 interacts with RNF12 and inhibits polyubiquitination. Instead, the authors should first ubiquitinate SMAD7, deplete ATP and then observed de-ubiquitination.

Response: In response to reviewer, we first ubiquitinated His-SMAD7 with RNF12 through incubation with E1, E2 (UbcH6) and ubiquitin *in vitro* for 3h at 37°C. The polyubiquitinated His-SMAD7 was immunoprecipitated by anti-SMAD7 antibody. The beads were washed and ATP was depleted then incubated with purified OTUD1 wt/CA for 60 mins. The assay mixtures were analyzed by anti-Ub, anti-SMAD7 and anti-OTUD1 antibodies (see new Figure 4H). We accordingly modify our description in the text.

3) The authors often say OTUD1 “inhibits” SMAD7 poly-ubiquitination, whereas it deubiquitinates SMAD7. Inhibiting ubiquitination is a completely different mechanism, which is generally not a DUB activity.

Response: Thank you for pointing this out. We corrected these mistakes and refer to it as “deubiquitinate”.

4) Figure 5A right lane. While wt-OTUD1 can completely cleave K48-linked –SMAD7 in the DMSO treated cells, it does not in the MG132 treated cells. Do the DUBS present at the proteasome have a role in de-ubiquitinating SMAD7?

Response: Yes, our results shown in Figure 5A indeed suggests that certain DUBs present at the proteasome might be responsible for a certain level of SMAD7 deubiquitination and those DUBs are suppressed by MG132. We have included this point into our discussion (highlighted in Page 20).

5) The authors use both “ubiquitination” and “ubiquitylation”. It is preferable to stick to any one of these terms.

Response: All the “ubiquitylation” have been changed into “ubiquitination”.

6) Correct spelling “ubiquitination” in line 259.

Response: It has been corrected.

7) Line 277, introduce a blank between “withSMURF2”.

Response: It has been corrected.

Reviewer #1:

Remarks to the Author:

The authors have addressed most of my concerns. However, they deleted the data on MII cells, which are not consistent with their hypothesis that Ras signaling failed to down-regulate OTUD1 in p53-wt cells. It would be nice if the authors examine the hypothesis using p53-wt breast cancer cells, for example, MCF7 cells.

Reviewer #2:

Remarks to the Author:

The authors have addressed my concerns. No further review.

Reviewer #4:

Remarks to the Author:

The authors have adequately responded to my comments. I recommend this manuscript for publication.

Manuscript number: NCOMMS-17-00755-T

Rebuttal to comments by the reviewers

Reviewer #1 (Remarks to the Author):

The authors have addressed most of my concerns. However, they deleted the data on MII cells, which are not consistent with their hypothesis that Ras signaling failed to down-regulate OTUD1 in p53-wt cells. It would be nice if the authors examine the hypothesis using p53-wt breast cancer cells, for example, MCF7 cells.

Response: In MCF7 cells, Ras could not significantly inhibit OTUD1 expression. This result had been shown in Fig. 8D.